# Tree Variational Autoencoders

**Laura Manduchi,**\* **Moritz Vandenhirtz,**\* **Alain Ryser, Julia E. Vogt**
Department of Computer Science
ETH Zurich
Switzerland

## Abstract

We propose *Tree Variational Autoencoder* (TreeVAE), a new generative hierarchical clustering model that learns a flexible tree-based posterior distribution over latent variables. TreeVAE hierarchically divides samples according to their intrinsic characteristics, shedding light on hidden structures in the data. It adapts its architecture to discover the optimal tree for encoding dependencies between latent variables. The proposed tree-based generative architecture enables lightweight conditional inference and improves generative performance by utilizing specialized leaf decoders. We show that TreeVAE uncovers underlying clusters in the data and finds meaningful hierarchical relations between the different groups on a variety of datasets, including real-world imaging data. We present empirically that TreeVAE provides a more competitive log-likelihood lower bound than the sequential counterparts. Finally, due to its generative nature, TreeVAE is able to generate new samples from the discovered clusters via conditional sampling.

## 1 Introduction

Discovering structure and hierarchies in the data has been a long-standing goal in machine learning (Bishop, 2006; Bengio et al., 2012; Jordan & Mitchell, 2015). Interpretable supervised methods, such as decision trees (Zhou & Feng, 2017; Tanno et al., 2019), have proven to be successful in unveiling hierarchical relationships within data. However, the expense of annotating large quantities of data has resulted in a surge of interest in unsupervised approaches (LeCun et al., 2015). Hierarchical clustering (Ward, 1963) offers an unsupervised path to find hidden groups in the data and their hierarchical relationship (R. J. G. B. Campello et al., 2015). Due to its versatility, interpretability, and ability to uncover meaningful patterns in complex data, hierarchical clustering has been widely used in a variety of applications, including phylogenetics (Sneath & Sokal, 1962), astrophysics (McConnachie et al., 2018), and federated learning (Briggs et al., 2020). Similar to how the human brain automatically categorizes and connects objects based on shared attributes, hierarchical clustering algorithms construct a dendrogram - a tree-like structure of clusters - that organizes data into nested groups based on their similarity. Despite its potential, hierarchical clustering has taken a step back in light of recent advances in self-supervised deep learning (Chen et al., 2020), and only a few deep learning based methods have been proposed in recent years (Goyal et al., 2017; Mautz et al., 2020).

Deep latent variable models (Kingma & Welling, 2019), a class of generative models, have emerged as powerful frameworks for unsupervised learning and they have been extensively used to uncover hidden structures in the data (Dilokthanakul et al., 2016; Manduchi et al., 2021). They leverage the flexibility of neural networks to capture complex patterns and generate meaningful representations of high-dimensional data. By incorporating latent variables, these models can uncover the underlying factors of variation of the data, making them a valuable tool for understanding and modeling complex data distributions. In recent years, a variety of deep generative methods have been proposed to incorporate more complex posterior distributions by modeling structural *sequential* dependencies

---

\*Equal contribution. Correspondence to {`laura.manduchi,moritz.vandenhirtz`}`@inf.ethz.ch`

37th Conference on Neural Information Processing Systems (NeurIPS 2023).

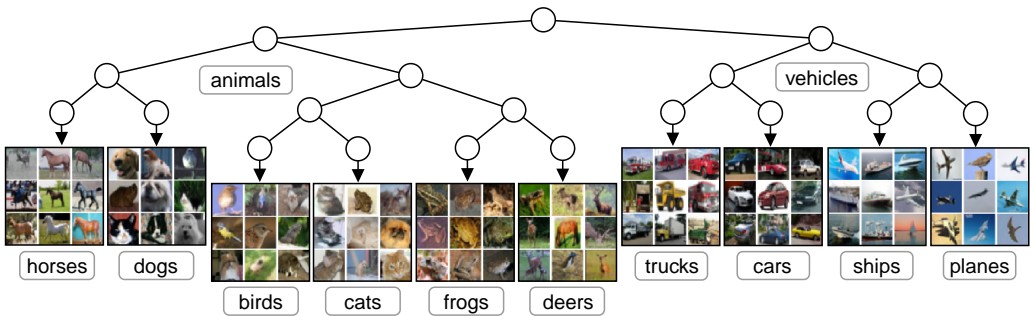

Figure 1: The hierarchical structure discovered by TreeVAE on the CIFAR-10 dataset. We display random subsets of images that are probabilistically assigned to each leaf of the tree.

between latent variables (Sønderby et al., 2016; He et al., 2018; Maaløe et al., 2019; Vahdat & Kautz, 2020a), thus offering different levels of abstraction for encoding the data distribution.

Our work advances the state-of-the-art in structured VAEs by combining the complementary strengths of hierarchical clustering algorithms and deep generative models. We propose TreeVAE[1], a novel tree-based generative model that encodes *hierarchical* dependencies between latent variables. We introduce a training procedure to learn the optimal tree structure to model the posterior distribution of latent variables. An example of a tree learned by TreeVAE is depicted in Fig. 1. Each edge and split are encoded by neural networks, while the circles depict latent variables. Each sample is associated with a probability distribution over *paths*. The resulting tree thus organizes the data into an interpretable hierarchical structure in an unsupervised fashion, optimizing the amount of shared information between samples. In CIFAR-10, for example, the method divides the vehicles and animals into two different subtrees and similar groups (such as planes and ships) share common ancestors.

**Our main contributions** are as follows: (*i*) We propose a novel, deep probabilistic approach to hierarchical clustering that learns the optimal generative binary tree to mimic the hierarchies present in the data. (*ii*) We provide a thorough empirical assessment of the proposed approach on MNIST, Fashion-MNIST, 20Newsgroups, and Omniglot. In particular, we show that TreeVAE (a) outperforms related work on deep hierarchical clustering, (b) discovers meaningful patterns in the data and their hierarchical relationships, and (c) achieves a more competitive log-likelihood lower bound compared to VAE and LadderVAE, its sequential counterpart. (*iii*) We propose an extension of TreeVAE that integrates contrastive learning into its tree structure. Relevant prior knowledge, expertise, or specific constraints can be incorporated into the generative model via augmentations, allowing for more accurate and contextually meaningful clustering. We test the contrastive version of TreeVAE on CIFAR-10, CIFAR-100, and CelebA, and we show that the proposed approach achieves competitive hierarchical clustering performance compared to the baselines.

## 2 TreeVAE

We propose TreeVAE, a novel deep generative model that learns a flexible tree-based posterior distribution over latent variables. Each sample travels through the tree from root to leaf in a probabilistic manner as TreeVAE learns sample-specific probability distributions of paths. As a result, the data is divided in a hierarchical fashion, with more refined concepts for deeper nodes in the tree. The proposed graphical model is depicted in Fig. 2. The inference and generative models share the same top-down tree structure, enabling interaction between the bottom-up and top-down architecture, similarly to Sønderby et al. (2016).

### 2.1 Model Formulation

Given $H$, the maximum depth of the tree, and a dataset $X$, the model is defined by three components that are learned during training:

---

[1]The code is publicly available at `https://github.com/lauramanduchi/treevae-pytorch`.

- the *global* structure of the binary tree $\mathcal{T}$, which specifies the set of nodes $\mathbb{V} = \{0, \ldots, V\}$, the set of leaves $\mathbb{L}$, where $\mathbb{L} \subset \mathbb{V}$, and the set of edges $\mathcal{E}$. See Fig. 1/4/5/6/7 for different examples of tree structures learned by the model.

- the *sample-specific* latent embeddings $\mathbf{z} = \{\mathbf{z}_0, \ldots, \mathbf{z}_V\}$, which are random variables assigned to each node in $\mathbb{V}$. Each embedding is characterized by a Gaussian distribution whose parameters are a function of the realization of the parent node. The dimensions of the latent embeddings are defined by their depth, with $\boldsymbol{z}_i \in \mathbb{R}^{h_{\mathrm{depth}(i)}}$ where $\mathrm{depth}(i)$ is the depth of the node $i$, and $h_{\mathrm{depth}(i)}$ is the embedding dimension for that depth.

- the *sample-specific* decisions $\mathbf{c} = \{\mathbf{c}_0, \ldots, \mathbf{c}_{V-|\mathbb{L}|}\}$, which are Bernoulli random variables defined by the probability of going to the right (or left) child of the underlying node. They take values $c_i \in \{0, 1\}$ for $i \in \mathbb{V} \setminus \mathbb{L}$, with $c_i = 0$ if the left child is selected. A decision path, $\mathcal{P}_l$, indicates the path from root to leaf given the tree $\mathcal{T}$ and is defined by the nodes in the path, e.g., in Fig. 2, $\mathcal{P}_l = \{0, 1, 4, 5\}$. The probability of $\mathcal{P}_l$ is the product of the probabilities of the decisions in the path.

The tree structure is shared across the entire dataset and is learned iteratively by growing the tree node-wise. The latent embeddings and the decision paths, on the other hand, are learned using variational inference by conditioning the model on the current tree structure. The generative/inference model and the learning objective conditioned on $\mathcal{T}$ are explained in Sec. 2.2/2.3/2.4 respectively, while in 2.5, we elaborate on the efficient growing procedure of the tree.

## 2.2 Generative Model

The generative process of TreeVAE for a given $\mathcal{T}$ is depicted in Fig. 2 (right). The generation of a new sample $x$ starts from the root. First, the latent embedding of the root node $\mathbf{z}_0$ is sampled from a standard Gaussian $p_\theta(\boldsymbol{z}_0) = \mathcal{N}(\boldsymbol{z}_0 \mid \mathbf{0}, \boldsymbol{I})$. Then, given the sampled $\boldsymbol{z}_0$, the decision of going to the left or the right node is sampled from a Bernoulli distribution $p(\mathbf{c}_0 \mid \boldsymbol{z}_0) = Ber(r_{p,0}(\boldsymbol{z}_0))$, where $\{r_{p,i} \mid i \in \mathbb{V} \setminus \mathbb{L}\}$ are functions parametrized by neural networks defined as *routers*, and cause the splits in Fig. 2. The subscript $p$ is used to indicate the parameters of the generative model. The latent embedding of the selected child, let us assume it is $\boldsymbol{z}_1$, is then sampled from a Gaussian distribution $p_\theta(\boldsymbol{z}_1 \mid \boldsymbol{z}_0) = \mathcal{N}(\mathbf{z}_1 \mid \mu_{p,1}(\boldsymbol{z}_0), \sigma_{p,1}^2(\boldsymbol{z}_0))$, where

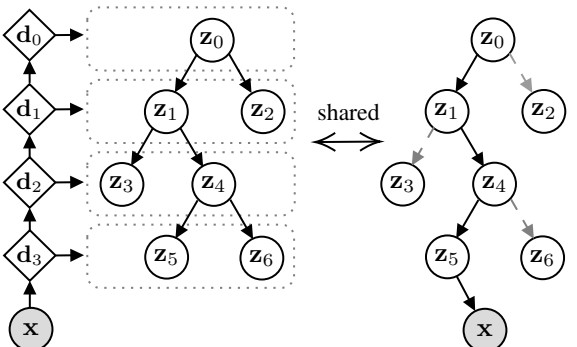

Figure 2: The proposed inference (left) and generative (right) models for TreeVAE. Circles are stochastic variables while diamonds are deterministic. The global topology of the tree is learned during training.

$\{\mu_{p,i}, \sigma_{p,i} \mid i \in \mathbb{V} \setminus \{0\}\}$ are functions parametrized by neural networks defined as *transformations*. They are indicated by the top-down arrows in Fig. 2. This process continues until a leaf is reached.

Let us define the set of latent variables selected by the path $\mathcal{P}_l$, which goes from the root to the leaf $l$, as $\mathbf{z}_{\mathcal{P}_l} = \{\mathbf{z}_i \mid i \in \mathcal{P}_l\}$, the parent node of the node $i$ as $pa(i)$, and $p(c_{pa(i) \to i} \mid \boldsymbol{z}_{pa(i)})$ the probability of going from $pa(i)$ to $i$. Note that the path $\mathcal{P}_l$ defines the sequence of decisions. The prior probability of the latent embeddings and the path given the tree $\mathcal{T}$ can be summarized as

$$p_\theta(\boldsymbol{z}_{\mathcal{P}_l}, \mathcal{P}_l) = p(\boldsymbol{z}_0) \prod_{i \in \mathcal{P}_l \setminus \{0\}} p(c_{pa(i) \to i} \mid \boldsymbol{z}_{pa(i)}) p(\boldsymbol{z}_i \mid \boldsymbol{z}_{pa(i)}). \tag{1}$$

Finally, $x$ is sampled from a distribution that is conditioned on the selected leaf. If we assume that $x$ is real-valued, then

$$p_\theta(\boldsymbol{x} \mid \boldsymbol{z}_{\mathcal{P}_l}, \mathcal{P}_l) = \mathcal{N}\left(\mathbf{x} \mid \mu_{x,l}(\boldsymbol{z}_l), \sigma_{x,l}^2(\boldsymbol{z}_l)\right), \tag{2}$$

where $\{\mu_{x,l}, \sigma_{x,l} \mid l \in \mathbb{L}\}$ are functions parametrized by leaf-specific neural networks defined as *decoders*.

## 2.3  Inference Model

The inference model is described by the variational posterior distribution of both the latent embeddings and the paths. It follows a similar structure as in the prior probability defined in (1), with the difference that the probability of the root and of the decisions are now conditioned on the sample $\boldsymbol{x}$:

$$q(\boldsymbol{z}_{\mathcal{P}_l}, \mathcal{P}_l \mid \boldsymbol{x}) = q(\boldsymbol{z}_0 \mid \boldsymbol{x}) \prod_{i \in \mathcal{P}_l \setminus \{0\}} q(c_{pa(i) \to i} \mid \boldsymbol{x}) q(\boldsymbol{z}_i \mid \boldsymbol{z}_{pa(i)}). \tag{3}$$

To compute the variational probability distribution of the latent embeddings $q(\boldsymbol{z}_0 \mid \boldsymbol{x})$ and $q(\boldsymbol{z}_i \mid \boldsymbol{z}_{pa(i)})$, where

$$q(\boldsymbol{z}_0 \mid \boldsymbol{x}) = \mathcal{N}\left(\boldsymbol{z}_0 \mid \mu_{q,0}(\boldsymbol{x}), \sigma_{q,0}^2(\boldsymbol{x})\right) \tag{4}$$

$$q_\phi\left(\mathbf{z}_i \mid \boldsymbol{z}_{pa(i)}\right) = \mathcal{N}\left(\mathbf{z}_i \mid \mu_{q,i}\left(\boldsymbol{z}_{pa(i)}\right), \sigma_{q,i}^2\left(\boldsymbol{z}_{pa(i)}\right)\right), \forall i \in \mathcal{P}_l, \tag{5}$$

we follow a similar approach to the one proposed by Sønderby et al. (2016). Note that we use the subscript $q$ to indicate the parameters of the inference model.

First, a deterministic bottom-up pass computes the node-specific approximate likelihood contributions

$$\mathbf{d}_h = \mathrm{MLP}\left(\mathbf{d}_{h+1}\right) \tag{6}$$

$$\hat{\boldsymbol{\mu}}_{q,i} = \mathrm{Linear}\left(\mathbf{d}_{depth(i)}\right), i \in \mathbb{V} \tag{7}$$

$$\hat{\boldsymbol{\sigma}}_{q,i}^2 = \mathrm{Softplus}\left(\mathrm{Linear}\left(\mathbf{d}_{depth(i)}\right)\right), i \in \mathbb{V}, \tag{8}$$

where $\mathbf{d}_H$ is parametrized by a domain-specific neural network defined as *encoder*, and $\mathrm{MLP}(\mathbf{d}_h)$ for $h \in \{1, \ldots, H\}$, indicated by the bottom-up arrows in Fig. 2, are neural networks, shared among the parameter predictors, $\hat{\mu}_{q,i}, \hat{\sigma}_{q,i}^2$, of the same depth. They are characterized by the same architecture as the *transformations* defined in Sec.2.2.

A stochastic downward pass then recursively computes the approximate posteriors defined as

$$\boldsymbol{\sigma}_{q,i}^2 = \frac{1}{\hat{\boldsymbol{\sigma}}_{q,i}^{-2} + \boldsymbol{\sigma}_{p,i}^{-2}}, \quad \boldsymbol{\mu}_{q,i} = \frac{\hat{\boldsymbol{\mu}}_{q,i}\hat{\boldsymbol{\sigma}}_{q,i}^{-2} + \boldsymbol{\mu}_{p,i}\boldsymbol{\sigma}_{p,i}^{-2}}{\hat{\boldsymbol{\sigma}}_{q,i}^{-2} + \boldsymbol{\sigma}_{p,i}^{-2}}, \tag{9}$$

where all operations are performed elementwise. Finally, the variational distributions of the decisions $q(c_i \mid \boldsymbol{x})$ are defined as

$$q(c_i \mid \boldsymbol{x}) = q(c_i \mid \mathbf{d}_{\mathrm{depth(i)}}) = Ber(r_{q,i}(\mathbf{d}_{\mathrm{depth(i)}})), \tag{10}$$

where $\{r_{q,i} \mid i \in \mathbb{V} \setminus \mathbb{L}\}$ are functions parametrized by neural networks and are characterized by the same architecture as the *routers* of the generative model defined in Sec. 2.2.

## 2.4  Evidence Lower Bound

The parameters of both the generative model (defined as $p$) and inference model (defined as $q$), consisting of the encoder ($\mu_{q,0}, \sigma_{q,0}$), the transformations ($\{(\mu_{p,i}, \sigma_{p,i}), (\mu_{q,i}, \sigma_{q,i}) \mid i \in \mathbb{V} \setminus \{0\}\}$), the decoders ($\{\mu_{x,l}, \sigma_{x,l} \mid l \in \mathbb{L}\}$) and the routers ($\{r_{p,i}, r_{q,i} \mid i \in \mathbb{V} \setminus \mathbb{L}\}$), are learned by maximizing the Evidence Lower Bound (ELBO) (Kingma & Welling, 2014; Rezende et al., 2014). Each leaf $l$ is associated with only one path $\mathcal{P}_l$, hence we can write the data likelihood conditioned on $\mathcal{T}$ as

$$p(\boldsymbol{x} \mid \mathcal{T}) = \sum_{l \in \mathbb{L}} \int_{\boldsymbol{z}_{\mathcal{P}_l}} p(\boldsymbol{x}, \boldsymbol{z}_{\mathcal{P}_l}, \mathcal{P}_l) = \sum_{l \in \mathbb{L}} \int_{\boldsymbol{z}_{\mathcal{P}_l}} p_\theta(\boldsymbol{z}_{\mathcal{P}_l}, \mathcal{P}_l) p_\theta(\boldsymbol{x} \mid \boldsymbol{z}_{\mathcal{P}_l}, \mathcal{P}_l). \tag{11}$$

We use variational inference to derive the ELBO of the log-likelihood:

$$\mathcal{L}(\boldsymbol{x} \mid \mathcal{T}) := \mathbb{E}_{q(\boldsymbol{z}_{\mathcal{P}_l}, \mathcal{P}_l \mid \boldsymbol{x})}[\log p(\boldsymbol{x} \mid \boldsymbol{z}_{\mathcal{P}_l}, \mathcal{P}_l)] - \mathrm{KL}\left(q\left(\boldsymbol{z}_{\mathcal{P}_l}, \mathcal{P}_l \mid \boldsymbol{x}\right) \| p\left(\boldsymbol{z}_{\mathcal{P}_l}, \mathcal{P}_l\right)\right). \tag{12}$$

The first term of the ELBO is the reconstruction term:

$$\mathcal{L}_{rec} = \mathbb{E}_{q(\boldsymbol{z}_{\mathcal{P}_l}, \mathcal{P}_l \mid \boldsymbol{x})}[\log p(\boldsymbol{x} \mid \boldsymbol{z}_{\mathcal{P}_l}, \mathcal{P}_l)] \tag{13}$$

$$= \sum_{l \in \mathbb{L}} \int_{\boldsymbol{z}_{\mathcal{P}_l}} q(\boldsymbol{z}_0 \mid \boldsymbol{x}) \prod_{i \in \mathcal{P}_l \setminus \{0\}} q(c_{pa(i) \to i} \mid \boldsymbol{x}) q(\boldsymbol{z}_i \mid \boldsymbol{z}_{pa(i)}) \log p(\boldsymbol{x} \mid \boldsymbol{z}_{\mathcal{P}_l}, \mathcal{P}_l) \tag{14}$$

$$\approx \frac{1}{M} \sum_{m=1}^{M} \sum_{l \in \mathbb{L}} P(l; \boldsymbol{c}) \log \mathcal{N}\left(\mathbf{x} \mid \mu_{x,l}\left(\boldsymbol{z}_l^{(m)}\right), \sigma_{x,l}^2\left(\boldsymbol{z}_l^{(m)}\right)\right), \tag{15}$$

$$P(i; \boldsymbol{c}) = \prod_{j \in \mathcal{P}_i \setminus \{0\}} q(c_{pa(j) \to j} \mid \boldsymbol{x}) \quad \text{for } i \in \mathbb{V}, \tag{16}$$

where $\mathcal{P}_i$ for $i \in \mathbb{V}$ is the path from root to node $i$, $P(i; \boldsymbol{c})$ is the probability of reaching node $i$, which is the product over the probabilities of the decisions in the path until $i$, $\boldsymbol{z}_l^{(m)}$ are the Monte Carlo (MC) samples, and $M$ the number of the MC samples. Intuitively, the reconstruction loss is the sum of the leaf-wise reconstruction losses weighted by the probabilities of reaching the respective leaf. Note that here we sum over all possible paths in the tree, which is equal to the number of leaves.

The second term of (12) is the Kullback–Leibler divergence (KL) between the prior and the variational posterior of the tree. It can be written as a sum of the KL of the root, the nodes, and the decisions:

$$\mathrm{KL}\left(q\left(\boldsymbol{z}_{\mathcal{P}_l}, \mathcal{P}_l \mid \boldsymbol{x}\right) \| p\left(\boldsymbol{z}_{\mathcal{P}_l}, \mathcal{P}_l\right)\right) = \mathrm{KL}_{root} + \mathrm{KL}_{nodes} + \mathrm{KL}_{decisions} \tag{17}$$

$$\mathrm{KL}_{root} = \mathrm{KL}(q(\boldsymbol{z}_0 \mid \boldsymbol{x}) \| p(\boldsymbol{z}_0)) \tag{18}$$

$$\mathrm{KL}_{nodes} \approx \frac{1}{M} \sum_{m=1}^{M} \sum_{i \in \mathbb{V} \setminus \{0\}} P(i; \boldsymbol{c}) \, \mathrm{KL}(q(\boldsymbol{z}_i^{(m)} \mid pa(\boldsymbol{z}_i^{(m)})) \| p(\boldsymbol{z}_i^{(m)} \mid pa(\boldsymbol{z}_i^{(m)}))) \tag{19}$$

$$\mathrm{KL}_{decisions} \approx \frac{1}{M} \sum_{m=1}^{M} \sum_{i \in \mathbb{V} \setminus \mathbb{L}} P(i; \boldsymbol{c}) \sum_{c_i \in \{0,1\}} q(c_i \mid \boldsymbol{x}) \log \left( \frac{q(c_i \mid \boldsymbol{x})}{p(c_i \mid \boldsymbol{z}_i^{(m)})} \right), \tag{20}$$

where $M$ is the number of MC samples. We refer to Appendix A for the full derivation. The $\mathrm{KL}_{root}$ is the KL between the standard Gaussian prior $p(\boldsymbol{z}_0)$ and the variational posterior of the root $q(\boldsymbol{z}_0 \mid \boldsymbol{x})$, thus enforcing the root to be compact. The $\mathrm{KL}_{nodes}$ is the sum of the node-specific KLs weighted by the probability of reaching their node $i$: $P(i; \boldsymbol{c})$. The node-specific KL of node $i$ is the KL between the two Gaussians $q(\boldsymbol{z}_i \mid pa(\boldsymbol{z}_i))$, $p(\boldsymbol{z}_i \mid pa(\boldsymbol{z}_i))$. Finally, the last term, $\mathrm{KL}_{decisions}$, is the weighted sum of all the KLs of the decisions, which are Bernoulli random variables, $KL(q(c_i \mid \boldsymbol{x}) \mid p(c_i \mid \boldsymbol{z}_i))) = \sum_{c_i \in \{0,1\}} q(c_i \mid \boldsymbol{x}) \log \left( \frac{q(c_i \mid \boldsymbol{x})}{p(c_i \mid \boldsymbol{z}_i)} \right)$. The hierarchical specification of the binary tree allows encoding highly expressive models while retaining the computational efficiency of fully factorized models. The computational complexity is described in Appendix A.2.

## 2.5 Growing The Tree

In the previous sections, we discussed the variational objective to learn the parameters of both the generative and the inference model given a defined tree structure $\mathcal{T}$. Here we discuss how to learn the structure of the binary tree $\mathcal{T}$. TreeVAE starts by training a tree composed of a root and two leaves, see Fig. 3 (left), for $N_t$ epochs by optimizing the ELBO. Once the model converged, a leaf is selected, e.g., $\boldsymbol{z}_1$ in Fig. 3, and two children are attached to it. The leaf selection criteria can vary depending on the application and can be determined by, e.g., the reconstruction loss or the ELBO. In our experiments, we chose to select the nodes with the maximum number of samples to retain balanced

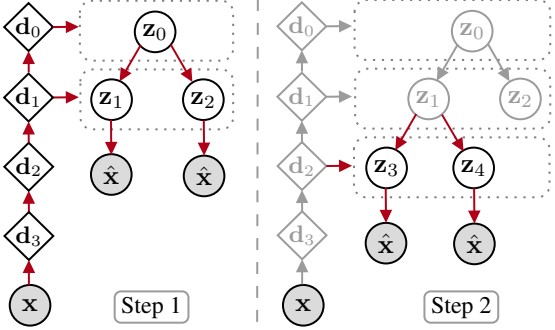

Figure 3: The first two steps of the growing process to learn the global structure of the tree during training. Highlighted in red are the trainable weights.

leaves. The sub-tree composed of the new leaves and the parent node is then trained for $N_t$ epochs by freezing the weights of the rest of the model, see Fig. 3 (right), resulting in computing the ELBO of the nodes of the subtree. For efficiency, the subtree is trained using only the subset of data that have a high probability (higher than a threshold $t$) of being assigned to the parent node. The process is repeated until the tree reaches its maximum capacity (defined by the maximum depth) or until a condition (such as a predefined maximum number of leaves) is met. The entire model is then fine-tuned for $N_f$ epochs by unfreezing all weights. During fine-tuning, the tree is pruned by removing empty branches (with the expected number of assigned samples lower than a threshold).

## 2.6 Integrating Prior Knowledge

Retrieving semantically meaningful clustering structures of real-world images is extremely challenging, as there are several underlying factors according to which the data can be clustered. Therefore, it

is often crucial to integrate domain knowledge that guides the model toward desirable cluster assignments. Thus, we propose an extension of TreeVAE where we integrate recent advances in contrastive learning (van den Oord et al., 2018; Chen et al., 2020; Li et al., 2021), whereby prior knowledge on data invariances can be encoded through augmentations. For a batch $\boldsymbol{X}$ with $N$ samples, we randomly augment every sample twice to obtain the augmented batch $\tilde{X}$ with $2N$ samples. For all positive pairs $(i, j)$ where $\tilde{\boldsymbol{x}}_i$ and $\tilde{\boldsymbol{x}}_j$ stem from the same original sample, we utilize the *NT-Xent* (Chen et al., 2020), which introduces losses $\ell_{i,j} = -\log \frac{\exp(s_{i,j}/\tau)}{\sum_{k=1}^{2N} \mathbb{1}_{[k \neq i]} \exp(s_{i,k}/\tau)}$, where $s_{i,j}$ denotes the cosine similarity between the representations of $\tilde{\boldsymbol{x}}_i$ and $\tilde{\boldsymbol{x}}_j$, and $\tau$ is a temperature parameter. We integrate $\ell_{i,j}$ in both the bottom-up and the routers of TreeVAE. In the bottom-up, similar to Chen et al. (2020), we compute $\ell_{i,j}$ on the projections $g_h(\mathbf{d}_h)$. For the routers, we directly compute the loss on the predicted probabilities $r_{q,i}(\mathbf{d}_h)$. Finally, we average the terms over all positive pairs and add them to the negative ELBO (12) in real-world image experiments. Implementation details can be found in Appendix E, while a loss ablation is shown in Appendix C.3.

## 3 Related Work

Deep latent variable models automatically learn structure from data by combining the flexibility of deep neural networks and the statistical foundations of generative models (Mattei & Frellsen, 2018). Variational autoencoders (VAEs) (Rezende et al., 2014; Kingma & Welling, 2014) are among the most used frameworks (Nasiri & Bepler, 2022; Bae et al., 2023; Bredell et al., 2023). A variety of works has been proposed to integrate more complex empirical prior distributions, thus reducing the gap between approximate and true posterior distributions (Ranganath et al., 2015; Webb et al., 2017; Klushyn et al., 2019). Among these, the most related to our work is the VAE-nCRP (Goyal et al., 2017; Shin et al., 2019) and the TMC-VAE (Vikram et al., 2018). Both works use Bayesian nonparametric hierarchical clustering based on the nested Chinese restaurant process (nCRP) prior (Blei et al., 2003), and on the time-marginalized coalescent (TMC). However, even if they allow more flexible prior distributions these models suffer from restrictive posterior distributions (Kingma et al., 2016).To overcome the above issue, deep hierarchical VAEs (Gregor et al., 2015; Kingma et al., 2016) have been proposed to employ structured approximate posteriors, which are composed of hierarchies of conditional stochastic variables that are connected *sequentially*. Among a variety of proposed methods (Vahdat & Kautz, 2020b; Falck et al., 2022; T. Z. Xiao & Bamler, 2023), Ladder VAE (Sønderby et al., 2016) is most related to TreeVAE. The authors propose to model the approximate posterior by combining a "bottom-up" recognition distribution with the "top-down" prior. Further extensions include BIVA (Maaløe et al., 2019), which introduces a bidirectional inference network, and GraphVAE (He et al., 2019), that introduces gated dependencies over a fixed number of latent variables. Contrary to the previous approaches, TreeVAE models a *tree-based* posterior distribution of latent variable, thus allowing hierarchical clustering of samples. For further work on hierarchical clustering and its supervised counterpart, decision trees, we refer to Appendix B.

## 4 Experimental Setup

**Datasets and Metrics:** We evaluate the clustering and generative performance of TreeVAE on MNIST (LeCun et al., 1998), Fashion-MNIST (H. Xiao et al., 2017), 20Newsgroups (Lang, 1995), Omniglot (Lake et al., 2015), and Omniglot-5, where only 5 vocabularies (Braille, Glagolitic, Cyrillic, Odia, and Bengali) are selected and used as true labels. We assess the hierarchical clustering performance by computing dendrogram purity (DP) and leaf purity (LP), as defined by (Kobren et al., 2017a) using the datasets labels, where we assume the number of true clusters is unknown. We also report standard clustering metrics, accuracy (ACC) and normalized mutual information (NMI), by setting the number of leaves for TreeVAE and for the baselines to the true number of clusters. In terms of generative performance, we compute the approximated true log-likelihood calculated using 1000 importance-weighted samples, together with the ELBO (12) and the reconstruction loss (16). We also perform hierarchical clustering experiments on real-world imaging data, namely CIFAR-10, CIFAR-100 (Krizhevsky & Hinton, 2009) with 20 superclasses as labels, and CelebA (Z. Liu et al., 2015) using the contrastive extension (Sec. 2.6). We refer to Appendix D for more dataset details.

**Baselines:** We compare the generative performance of TreeVAE to the VAE (Rezende et al., 2014; Kingma & Welling, 2014), its non-hierarchical counterpart, and the LadderVAE (Sønderby et al., 2016), its sequential counterpart. For a fair comparison, all methods share the same architecture and

Table 1: Test set hierarchical clustering performances (%) of TreeVAE compared with baselines. Means and standard deviations are computed across 10 runs with different random model initialization. The star "*" indicates real-world image datasets on which contrastive approaches were applied.

| Dataset | Method | DP | LP | ACC | NMI |
|---|---|---|---|---|---|
| MNIST | Agg | $63.7_{\pm 0.0}$ | $78.6_{\pm 0.0}$ | $69.5_{\pm 0.0}$ | $71.1_{\pm 0.0}$ |
| | VAE + Agg | $79.9_{\pm 2.2}$ | $90.8_{\pm 1.4}$ | $86.6_{\pm 4.9}$ | $81.6_{\pm 2.0}$ |
| | LadderVAE + Agg | $81.6_{\pm 3.9}$ | $90.9_{\pm 2.5}$ | $80.3_{\pm 5.6}$ | $82.0_{\pm 2.1}$ |
| | DeepECT | $74.6_{\pm 5.9}$ | $90.7_{\pm 3.2}$ | $74.9_{\pm 6.2}$ | $76.7_{\pm 4.2}$ |
| | TreeVAE (ours) | $\mathbf{87.9}_{\pm 4.9}$ | $\mathbf{96.0}_{\pm 1.9}$ | $\mathbf{90.2}_{\pm 7.5}$ | $\mathbf{90.0}_{\pm 4.6}$ |
| Fashion | Agg | $45.0_{\pm 0.0}$ | $67.6_{\pm 0.0}$ | $51.3_{\pm 0.0}$ | $52.6_{\pm 0.0}$ |
| | VAE + Agg | $44.3_{\pm 2.5}$ | $65.9_{\pm 2.3}$ | $54.9_{\pm 4.4}$ | $56.1_{\pm 3.2}$ |
| | LadderVAE + Agg | $49.5_{\pm 2.3}$ | $67.6_{\pm 1.2}$ | $55.9_{\pm 3.0}$ | $60.7_{\pm 1.4}$ |
| | DeepECT | $44.9_{\pm 3.3}$ | $67.8_{\pm 1.4}$ | $51.8_{\pm 5.7}$ | $57.7_{\pm 3.7}$ |
| | TreeVAE (ours) | $\mathbf{54.4}_{\pm 2.4}$ | $\mathbf{71.4}_{\pm 2.0}$ | $\mathbf{63.6}_{\pm 3.3}$ | $\mathbf{64.7}_{\pm 1.4}$ |
| 20Newsgroups | Agg | $13.1_{\pm 0.0}$ | $30.8_{\pm 0.0}$ | $26.1_{\pm 0.0}$ | $27.5_{\pm 0.0}$ |
| | VAE + Agg | $7.1_{\pm 0.3}$ | $18.1_{\pm 0.5}$ | $15.2_{\pm 0.4}$ | $11.6_{\pm 0.3}$ |
| | LadderVAE + Agg | $9.0_{\pm 0.2}$ | $20.0_{\pm 0.7}$ | $17.4_{\pm 0.9}$ | $17.8_{\pm 0.6}$ |
| | DeepECT | $9.3_{\pm 1.8}$ | $17.2_{\pm 3.8}$ | $15.6_{\pm 3.0}$ | $18.1_{\pm 4.1}$ |
| | TreeVAE (ours) | $\mathbf{17.5}_{\pm 1.5}$ | $\mathbf{38.4}_{\pm 1.6}$ | $\mathbf{32.8}_{\pm 2.3}$ | $\mathbf{34.4}_{\pm 1.5}$ |
| Omniglot-5 | Agg | $41.4_{\pm 0.0}$ | $63.7_{\pm 0.0}$ | $53.2_{\pm 0.0}$ | $33.3_{\pm 0.0}$ |
| | VAE + Agg | $46.3_{\pm 2.3}$ | $68.1_{\pm 1.6}$ | $52.9_{\pm 4.2}$ | $34.4_{\pm 2.9}$ |
| | LadderVAE + Agg | $49.8_{\pm 3.9}$ | $71.3_{\pm 2.0}$ | $59.6_{\pm 4.9}$ | $44.2_{\pm 4.7}$ |
| | DeepECT | $33.3_{\pm 2.5}$ | $55.1_{\pm 2.8}$ | $41.1_{\pm 4.2}$ | $23.5_{\pm 4.3}$ |
| | TreeVAE (ours) | $\mathbf{58.8}_{\pm 4.0}$ | $\mathbf{77.7}_{\pm 3.9}$ | $\mathbf{63.9}_{\pm 7.0}$ | $\mathbf{50.0}_{\pm 5.9}$ |
| CIFAR-10* | VAE + Agg | $10.54_{\pm 0.12}$ | $16.33_{\pm 0.15}$ | $14.43_{\pm 0.19}$ | $1.86_{\pm 1.66}$ |
| | LadderVAE + Agg | $12.81_{\pm 0.20}$ | $25.37_{\pm 0.62}$ | $19.29_{\pm 0.60}$ | $7.41_{\pm 0.42}$ |
| | DeepECT | $10.01_{\pm 0.02}$ | $10.30_{\pm 0.40}$ | $10.31_{\pm 0.39}$ | $0.18_{\pm 0.10}$ |
| | TreeVAE (ours) | $\mathbf{35.30}_{\pm 1.15}$ | $\mathbf{53.85}_{\pm 1.23}$ | $\mathbf{52.98}_{\pm 1.34}$ | $\mathbf{41.44}_{\pm 1.13}$ |
| CIFAR-100* | VAE + Agg | $5.27_{\pm 0.02}$ | $9.86_{\pm 0.19}$ | $8.82_{\pm 0.11}$ | $2.46_{\pm 0.10}$ |
| | LadderVAE + Agg | $6.36_{\pm 0.07}$ | $16.08_{\pm 0.28}$ | $14.01_{\pm 0.41}$ | $8.99_{\pm 0.41}$ |
| | DeepECT | $5.28_{\pm 0.18}$ | $6.97_{\pm 0.69}$ | $6.97_{\pm 0.69}$ | $1.71_{\pm 0.86}$ |
| | TreeVAE (ours) | $\mathbf{10.44}_{\pm 0.38}$ | $\mathbf{24.16}_{\pm 0.65}$ | $\mathbf{21.82}_{\pm 0.77}$ | $\mathbf{17.80}_{\pm 0.42}$ |

hyperparameters whenever possible. We compare TreeVAE to non-generative hierarchical clustering baselines for which the code was publicly available: Ward's minimum variance agglomerative clustering (Agg) (Ward, 1963; Murtagh & Legendre, 2014), and the DeepECT (Mautz et al., 2020). We propose two additional baselines, where we perform Ward's agglomerative clustering on the latent space of the VAE (VAE + Agg) and of the last layer of the LadderVAE (LadderVAE + Agg). For the contrastive clustering experiments, we apply a contrastive loss similar to TreeVAE to the VAE and the LadderVAE, while for DeepECT we use the contrastive loss proposed by the authors.

**Implementation Details:** While we believe that more complex architectures could have a substantial impact on the performance of TreeVAE, we choose to employ rather simple settings to validate the proposed approach. We set the dimension of all latent embeddings $\mathbf{z} = \{\mathbf{z}_0, \ldots, \mathbf{z}_V\}$ to 8 for MNIST, Fashion, and Omniglot, to 4 for 20Newsgroups, and to 64 for CIFAR-10, CIFAR-100, and CelebA. The maximum depth of the tree is set to 6 for all datasets, except 20Newsgroups where we increased the depth to 7 to capture more clusters. To compute DP and LP, we allow the tree to grow to a maximum of 30 leaves for 20Newsgroups and CIFAR-100, and 20 for the rest, while for ACC and NMI we fix the number of leaves to the number of true classes. The transformations consist of one-layer MLPs of size 128 and the routers of two-layers of size 128 for all datasets except for the real-world imaging data where we slightly increase the MLP complexity to 512. Finally, the encoder and decoders consist of simple CNNs and MLPs. The trees are trained for $N_t = 150$ epochs at each growth step, and the final tree is finetuned for $N_f = 200$ epochs. For the real-world imaging experiments, we set the weight of the contrastive loss to 100. See Appendix E for additional details.

Table 2: Test set generative performances of TreeVAE with 10 leaves compared with baselines. Means and standard deviations are computed across 10 runs with different random model initialization.

| Dataset | Method | LL | RL | ELBO |
|---|---|---|---|---|
| MNIST | VAE | $-101.9_{\pm 0.2}$ | $87.2_{\pm 0.3}$ | $-104.6_{\pm 0.3}$ |
| | LadderVAE | $-99.9_{\pm 0.5}$ | $87.8_{\pm 0.7}$ | $-103.2_{\pm 0.7}$ |
| | TreeVAE (ours) | $\mathbf{-92.9}_{\pm 0.2}$ | $\mathbf{80.3}_{\pm 0.2}$ | $\mathbf{-96.8}_{\pm 0.2}$ |
| Fashion | VAE | $-242.2_{\pm 0.2}$ | $231.7_{\pm 0.5}$ | $-245.4_{\pm 0.5}$ |
| | LadderVAE | $-239.4_{\pm 0.5}$ | $231.5_{\pm 0.6}$ | $-243.0_{\pm 0.6}$ |
| | TreeVAE (Ours) | $\mathbf{-234.7}_{\pm 0.1}$ | $\mathbf{226.5}_{\pm 0.3}$ | $\mathbf{-239.2}_{\pm 0.4}$ |
| 20Newsgroups | VAE | $\mathbf{-44.26}_{\pm 0.01}$ | $45.52_{\pm 0.03}$ | $\mathbf{-44.61}_{\pm 0.01}$ |
| | LadderVAE | $-44.30_{\pm 0.03}$ | $\mathbf{43.52}_{\pm 0.03}$ | $\mathbf{-44.62}_{\pm 0.02}$ |
| | TreeVAE (Ours) | $-51.67_{\pm 0.59}$ | $45.83_{\pm 0.36}$ | $-52.79_{\pm 0.66}$ |
| Omniglot | VAE | $-115.3_{\pm 0.3}$ | $101.6_{\pm 0.3}$ | $-118.2_{\pm 0.3}$ |
| | LadderVAE | $-113.1_{\pm 0.5}$ | $100.7_{\pm 0.7}$ | $-117.5_{\pm 0.6}$ |
| | TreeVAE (Ours) | $\mathbf{-110.4}_{\pm 0.5}$ | $\mathbf{96.9}_{\pm 0.5}$ | $\mathbf{-114.6}_{\pm 0.4}$ |

## 5 Results

**Hierarchical Clustering Results**   Table 1 shows the quantitative hierarchical clustering results averaged across 10 seeds. First, we assume the true number of clusters is *unknown* and report DP and LP. Second, we assume we have access to the true number of clusters $K$ and compute ACC and NMI. As can be seen, TreeVAE outperforms the baselines in both experiments. This suggests that the proposed approach successfully builds an optimal tree based on the data's intrinsic characteristics. Among the different baselines, agglomerative clustering using Ward's method (Agg) trained on the last layer of LadderVAE shows competitive performances. To the best of our knowledge, we are the first to report these results. It is noteworthy to observe that it consistently improves over VAE + Agg, indicating that the last layer of LadderVAE captures more cluster information than the VAE.

**Generative Results**   In Table 2, we evaluate the generative performance of the proposed approach, TreeVAE, compared to the VAE, its non-hierarchical counterpart, and LadderVAE, its sequential counterpart. TreeVAE outperforms the baselines on the majority of datasets, indicating that the proposed ELBO (12) can achieve a tighter lower bound of the log-likelihood. The most notable improvement appears to be reflected in the reconstruction loss, showing the advantage of using cluster-specialized decoders. However, this improvement comes at the expense of a larger neural network architecture and an increase in the number of parameters (as TreeVAE has $L$ decoders). While this requires more computational resources at training time, during deployment the tree structure of TreeVAE permits lightweight inference through conditional sampling, thus matching the inference time of LadderVAE. It is also worth mentioning that results differ from (Sønderby et al., 2016) as we adapt their architecture to match our experimental setting and consequently use smaller latent dimensionality. Finally, we notice that more complex methods are prone to overfitting on the 20Newsgroups dataset, hence the best performances are achieved by the VAE.

**Real-world Imaging Data & Contrastive Learning**   Clustering real-world imaging data is extremely difficult as there are endless possibilities of how the data can be partitioned (such as the colors, the landscape, etc). We therefore inject prior information through augmentations to guide TreeVAE and the baselines to semantically meaningful splits. Table 1 (bottom) shows the hierarchical clustering performance of TreeVAE and its baselines, all employing contrastive learning, on CIFAR-10 and CIFAR-100. We observe that DeepECT struggles in separating the data as their contrastive approach leads to all samples falling into the same leaf. In Table 3, we present the leaf-frequencies of various face attributes using the tree learned by TreeVAE. For all datasets, TreeVAE is able to group the data into contextually meaningful hierarchies and groups, evident from its superior performance compared to the baselines and from the distinct attribute frequencies in the leaves and subtrees.

**Discovery of Hierarchies**   In addition to solely clustering data, TreeVAE is able to discover meaningful hierarchical relations between the clusters, thus allowing for more insights into the

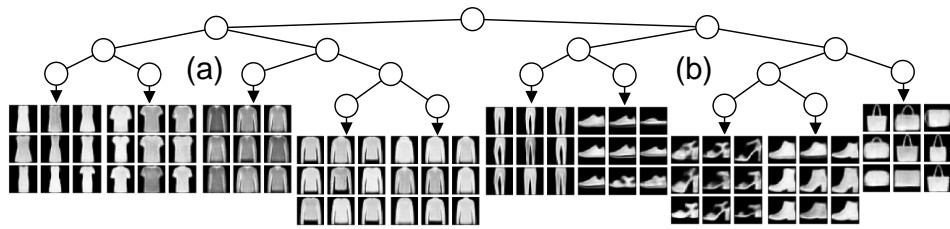

Figure 4: Hierarchical structures learned by TreeVAE on Fashion. Subtree (a) encodes tops, while (b) encodes shoes, purses, and pants.

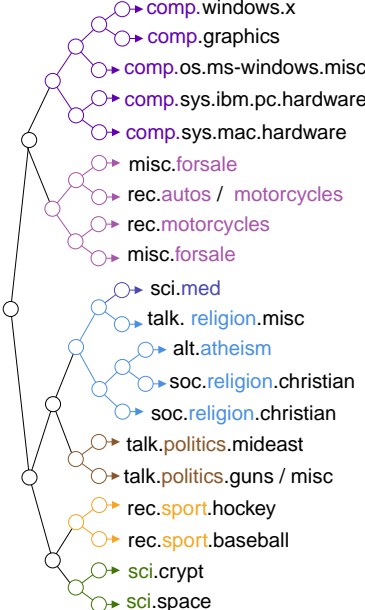

Figure 5: Hierarchical structure learned by TreeVAE on 20News-groups.

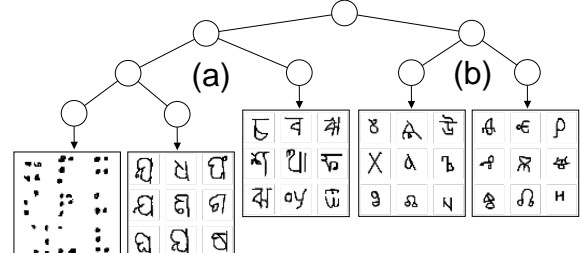

Figure 6: Hierarchical structures learned by TreeVAE on Omniglot-5. Subtree (a) learns a hierarchy over Braille and the Indian alphabets, while (b) groups Slavic alphabets.

| Attribute | 1 | 2 | 3 | 4 | 5 | 6 | 7 | 8 |
|---|---|---|---|---|---|---|---|---|
| Female | 97.2 | 55.0 | 97.7 | 86.6 | 23.1 | 30.7 | 46.6 | 43.7 |
| Bangs | 1.6 | 1.2 | 24.1 | 61.7 | 3.4 | 11.1 | 9.1 | 11.4 |
| Blonde | 1.1 | 3.7 | 66.7 | 2.2 | 5.8 | 2.6 | 26.1 | 7.1 |
| Makeup | 75.7 | 43.4 | 76.6 | 59.7 | 15.0 | 12.4 | 16.3 | 12.8 |
| Smiling | 54.3 | 66.6 | 66.4 | 51.2 | 54.7 | 42.4 | 37.3 | 22.4 |
| Hair Loss | 3.6 | 17.8 | 3.0 | 0.2 | 18.9 | 6.9 | 21.2 | 10.6 |
| Beard | 1.1 | 20.6 | 0.4 | 3.7 | 39.5 | 36.5 | 21.3 | 21.4 |

Table 3: We present the frequency (in %) of selected attributes for each leaf of TreeVAE with eight leaves in CelebA.

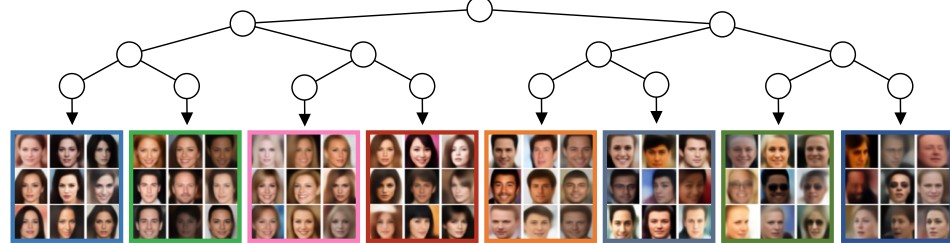

Figure 7: Hierarchical structure learned by TreeVAE with eight leaves on the CelebA dataset with generated images through conditional sampling. Generally, most females are in the left subtree, while most males are in the right subtree. We observe that leaf 1 is associated with dark-haired females, leaf 2 with smiling, dark-haired individuals, leaf 3 with blonde females, leaf 4 with bangs, leaf 7 with a receding hairline, and leaf 8 with non-smiling people. See Table 3 for further details.

dataset. In the introductory Fig. 1, 5, and 6, we present the hierarchical structures learned by TreeVAE, while in Fig. 4 and 7, we additionally display conditional cluster generations from the leaf-specific decoders. In Fig. 4, TreeVAE separates the fashion items into two subtrees, one containing shoes and bags, and the other containing the tops, which are further refined into long and short sleeves. In Fig. 5, we depict the most prevalent ground-truth topic label in each leaf. TreeVAE learns to

separate technological and societal subjects and discovers semantically meaningful subtrees. In Fig. 6, TreeVAE learns to split alphabets into Indian (Odia and Bengali) and Slavic (Glagolitic and Cyrillic) subtrees, while Braille is grouped with the Indian languages due to similar circle-like structures. For CelebA, Fig. 7 and Table 3, the resulting tree separates genders in the root split. Females (left) are further divided by hair color and hairstyle (bangs). Males (right) are further divided by smile intensity, beard, hair loss, and age. In Fig. 8 and Appendix C we show how TreeVAE can additionally be used to sample unconditional generations for all clusters simultaneously, by sampling from the root and propagating through the entire tree. The generations differ across the leaves by their cluster-specific features, whereas cluster-independent properties are retained across all generations.

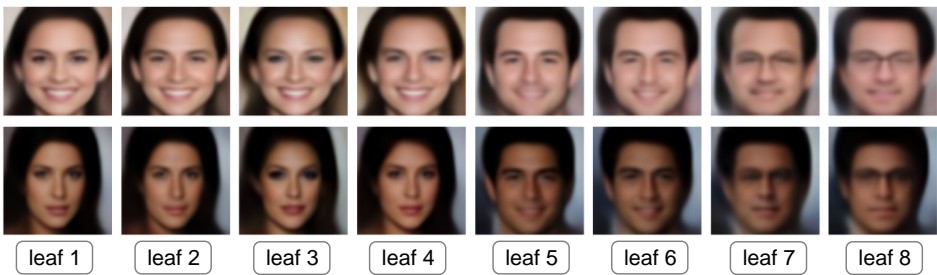

Figure 8: Selected unconditional generations of CelebA. One row corresponds to one sample from the root, for which we depict the visualizations obtained from the 8 leaf-decoders. The overall face shape, skin color, and face orientation are retained among leaves from the same row, while several properties (such as make-up, beard, mustache, glasses, and hair) vary across the different leaves.

## 6    Conclusion

In this paper, we introduced TreeVAE, a new generative method that leverages a tree-based posterior distribution of latent variables to capture the hierarchical structures present in the data. TreeVAE optimizes the balance between shared and specialized architecture, enhancing the learning and adaptation capabilities of generative models. Empirically, we showed that our model offers a substantial improvement in hierarchical clustering performance compared to the related work, while also providing a tighter lower bound to the log-likelihood of the data. We presented qualitatively how the hierarchical structures learned by TreeVAE enable a more comprehensive understanding of the data, thereby facilitating enhanced analysis, interpretation, and decision-making. Our findings highlight the versatility of the proposed approach, which we believe to hold significant potential for unsupervised representation learning, paving the way for exciting advancements in the field.

**Limitations & Future Work:**  Currently, TreeVAE uses a simple heuristic on which node to split that might not work on datasets with unbalanced clusters. Additionally, the contrastive losses on the routers encourage balanced clusters. Thus, more research is necessary to convert the heuristics to data-driven approaches. While deep latent variable models, such as VAEs, provide a framework for modeling explicit relationships through graphical structures, they often exhibit poor performance on synthetic image generation. However, more complex architectural design (Vahdat & Kautz, 2020a) or recent advancement in diffusion latent models (Rombach et al., 2021) present potential solutions to enhance image quality generation, thus striking an optimal balance between generating high-quality images and capturing meaningful representations.

## Acknowledgments and Disclosure of Funding

We thank Thomas M. Sutter for the insightful discussions throughout the project, Jorge da Silva Gonçalves for providing interpretable visualizations of the TreeVAE model, and Gabriele Manduchi for the valuable feedback on the notation of the ELBO. LM is supported by the SDSC PhD Fellowship #1-001568-037. MV is supported by the Swiss State Secretariat for Education, Research and Innovation (SERI) under contract number MB22.00047. AR is supported by the StimuLoop grant #1-007811-002 and the Vontobel Foundation.

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

# A Evidence Lower Bound

In this section, we provide a closer look at the loss function of TreeVAE. We focus on the derivations of the Kullback-Leibler divergence term of the Evidence Lower Bound and provide an interpretable factorization. Furthermore, we address the computational complexity, thus offering an in-depth understanding of the loss function, its practical implications, and the trade-offs involved in its computation.

## A.1 ELBO Derivations

In this section, we derive the KL loss (17) of the ELBO (12), which is the Kullback–Leibler divergence (KL) between the prior and the variational posterior of TreeVAE. Additionally, we give details about the underlying distributional assumptions for computing the reconstruction loss.

Let us define $\mathcal{P}_l$ the decision path from root $0$ to leaf $l$, $L$ is the number of leaves, which is equal to the number of paths in $\mathcal{T}$, $\mathbf{z}_{\mathcal{P}_l} = \{\mathbf{z}_i \mid i \in \mathcal{P}_l\}$ the set of latent variables selected by the path $\mathcal{P}_l$, the parent node of the node $i$ as $pa(i)$, $p(c_{pa(i)\to i} \mid \mathbf{z}_{pa(i)})$ the probability of going from $pa(i)$ to $i$. For example, if we consider the path in Fig. 2 (right) we will observe $c_0 = 0$, $c_1 = 1$, and $c_4 = 0$, where $c_i = 0$ means the model selects the left child of node $i$. The KL loss can be expanded using Eq. 1/3:

$$\mathrm{KL}\left(q\left(\mathbf{z}_{\mathcal{P}_l}, \mathcal{P}_l \mid \mathbf{x}\right) \| p\left(\mathbf{z}_{\mathcal{P}_l}, \mathcal{P}_l\right)\right) \tag{21}$$

$$= \mathrm{KL}\left(q(\mathbf{z}_0 \mid \mathbf{x}) \prod_{i \in \mathcal{P}_l \setminus \{0\}} q(c_{pa(i)\to i} \mid \mathbf{x}) q(\mathbf{z}_i \mid \mathbf{z}_{pa(i)}) \right.$$
$$\left. \left\| p(\mathbf{z}_0) \prod_{i \in \mathcal{P}_l \setminus \{0\}} p(c_{pa(i)\to i} \mid \mathbf{z}_{pa(i)}) p(\mathbf{z}_i \mid \mathbf{z}_{pa(i)})\right)\right. \tag{22}$$

$$= \sum_{l \in \mathbb{L}} \int_{\mathbf{z}_{\mathcal{P}_l}} q(\mathbf{z}_0 \mid \mathbf{x}) \prod_{i \in \mathcal{P}_l \setminus \{0\}} q(c_{pa(i)\to i} \mid \mathbf{x}) q(\mathbf{z}_i \mid \mathbf{z}_{pa(i)})$$
$$\times \log\left(\frac{q(\mathbf{z}_0 \mid \mathbf{x}) \prod_{j \in \mathcal{P}_l \setminus \{0\}} q(c_{pa(j)\to j} \mid \mathbf{x}) q(\mathbf{z}_j \mid \mathbf{z}_{pa(j)})}{p(\mathbf{z}_0) \prod_{k \in \mathcal{P}_l \setminus \{0\}} p(c_{pa(k)\to k} \mid \mathbf{z}_{pa(k)}) p(\mathbf{z}_k \mid \mathbf{z}_{pa(k)})}\right) \tag{23}$$

$$= \sum_{l \in \mathbb{L}} \int_{\mathbf{z}_{\mathcal{P}_l}} q(\mathbf{z}_0 \mid \mathbf{x}) \prod_{i \in \mathcal{P}_l \setminus \{0\}} q(c_{pa(i)\to i} \mid \mathbf{x}) q(\mathbf{z}_i \mid \mathbf{z}_{pa(i)}) \log\left(\frac{q(\mathbf{z}_0 \mid \mathbf{x})}{p(\mathbf{z}_0)}\right) \tag{24}$$

$$+ \sum_{l \in \mathbb{L}} \int_{\mathbf{z}_{\mathcal{P}_l}} q(\mathbf{z}_0 \mid \mathbf{x}) \prod_{i \in \mathcal{P}_l \setminus \{0\}} q(c_{pa(i)\to i} \mid \mathbf{x}) q(\mathbf{z}_i \mid \mathbf{z}_{pa(i)}) \log\left(\frac{\prod_{j \in \mathcal{P}_l \setminus \{0\}} q(c_{pa(j)\to j} \mid \mathbf{x})}{\prod_{k \in \mathcal{P}_l \setminus \{0\}} p(c_{pa(k)\to k} \mid \mathbf{z}_{pa(k)})}\right) \tag{25}$$

$$+ \sum_{l \in \mathbb{L}} \int_{\mathbf{z}_{\mathcal{P}_l}} q(\mathbf{z}_0 \mid \mathbf{x}) \prod_{i \in \mathcal{P}_l \setminus \{0\}} q(c_{pa(i)\to i} \mid \mathbf{x}) q(\mathbf{z}_i \mid \mathbf{z}_{pa(i)}) \log\left(\frac{\prod_{j \in \mathcal{P}_l \setminus \{0\}} q(\mathbf{z}_j \mid \mathbf{z}_{pa(j)})}{\prod_{k \in \mathcal{P}_l \setminus \{0\}} p(\mathbf{z}_k \mid \mathbf{z}_{pa(k)})}\right). \tag{26}$$

In the following, we will simplify each of the three terms 24, 25, and 26 separately.

### A.1.1 KL Root

The term (24) corresponds to the KL of the root node. We can integrate out all the latent variables $\mathbf{z}_i$ for $i \neq 0$ and all decisions $c_i$. The first term can be then written as follows:

$$\mathrm{KL}_{root} = \sum_{i \in \{1,2\}} \int_{\mathbf{z}_0} q(\mathbf{z}_0 \mid \mathbf{x}) q(c_{0\to i} \mid \mathbf{z}_0) \log\left(\frac{q(\mathbf{z}_0 \mid \mathbf{x})}{p(\mathbf{z}_0)}\right) \tag{27}$$

$$= \int_{\mathbf{z}_0} q(\mathbf{z}_0 \mid \mathbf{x}) \left[\sum_{i \in \{1,2\}} q(c_{0\to i} \mid \mathbf{z}_0)\right] \log\left(\frac{q(\mathbf{z}_0 \mid \mathbf{x})}{p(\mathbf{z}_0)}\right) \tag{28}$$

$$= \int_{\mathbf{z}_0} q(\mathbf{z}_0 \mid \mathbf{x}) \left[q(\mathrm{c}_0 = 0 \mid \mathbf{z}_0) + q(\mathrm{c}_0 = 1 \mid \mathbf{z}_0)\right] \log\left(\frac{q(\mathbf{z}_0 \mid \mathbf{x})}{p(\mathbf{z}_0)}\right) \tag{29}$$

$$= \int_{\mathbf{z}_0} q(\mathbf{z}_0 \mid \mathbf{x}) \log\left(\frac{q(\mathbf{z}_0 \mid \mathbf{x})}{p(\mathbf{z}_0)}\right) = \mathrm{KL}\left(q(\mathbf{z}_0 \mid \mathbf{x}) \| p(\mathbf{z}_0)\right), \tag{30}$$

where $q(c_0 = 0 \mid \boldsymbol{z}_0) + q(c_0 = 1 \mid \boldsymbol{z}_0) = 1$ and $\mathrm{KL}\left(q(\boldsymbol{z}_0 \mid \boldsymbol{x}) \| p(\boldsymbol{z}_0)\right)$ is the KL between two Gaussians, which can be computed analytically.

### A.1.2 KL Decisions

The second term (25) corresponds to the KL of the decisions. We can pull out the product from the log, yielding

$$
\mathrm{KL}_{decisions} = \sum_{l \in \mathbb{L}} \int_{\boldsymbol{z}_{\mathcal{P}_l}} q(\boldsymbol{z}_0 \mid \boldsymbol{x}) \prod_{i \in \mathcal{P}_l \backslash \{0\}} q(c_{pa(i) \to i} \mid \boldsymbol{x}) q(\boldsymbol{z}_i \mid \boldsymbol{z}_{pa(i)})
$$

$$
\times \log \left( \prod_{j \in \mathcal{P}_l \backslash \{0\}} \frac{q(c_{pa(j) \to j} \mid \boldsymbol{x})}{p(c_{pa(j) \to j} \mid \boldsymbol{z}_{pa(j)})} \right) \tag{31}
$$

$$
= \sum_{l \in \mathbb{L}} \int_{\boldsymbol{z}_{\mathcal{P}_l}} \sum_{j \in \mathcal{P}_l \backslash \{0\}} q(\boldsymbol{z}_0 \mid \boldsymbol{x}) \prod_{i \in \mathcal{P}_l \backslash \{0\}} q(c_{pa(i) \to i} \mid \boldsymbol{x}) q(\boldsymbol{z}_i \mid \boldsymbol{z}_{pa(i)}) \log \left( \frac{q(c_{pa(j) \to j} \mid \boldsymbol{x})}{p(c_{pa(j) \to j} \mid \boldsymbol{z}_{pa(j)})} \right) \tag{32}
$$

Let us define as $\mathcal{P}_{l \in j}$ all paths that go through node $j$, as $\mathcal{P}_{\leq j}$ (denoted as $\mathcal{P}_j$ in the main text for brevity) the unique path that ends in the node $j$, and as $\mathcal{P}_{>j}$ all the possible paths that start from the node $j$ and continue to a leaf $l \in \mathbb{L}$. Similarly, let us define as $\boldsymbol{z}_{\leq j}$ all the latent embeddings that are contained in the path from the root to node $j$ and as $\boldsymbol{z}_{>j}$ all the latent embeddings of the nodes $i > j$ that can be reached from node $j$.

To factorize the above equation, we first change from a pathwise view to a nodewise view. Instead of summing over all possible leaves in the tree ($\sum_{l \in \mathbb{L}}$) and then over each contained node ($\sum_{j \in \mathcal{P}_l \backslash \{0\}}$), we sum over all nodes ($\sum_{j \in \mathbb{V} \backslash \{0\}}$) and then over each path that leads through the selected node ($\sum_{\mathcal{P}_{l \in j}}$).

$$
\sum_{l \in \mathbb{L}} \int_{\boldsymbol{z}_{\mathcal{P}_l}} \sum_{j \in \mathcal{P}_l \backslash \{0\}} q(\boldsymbol{z}_0 \mid \boldsymbol{x}) \prod_{i \in \mathcal{P}_l \backslash \{0\}} q(c_{pa(i) \to i} \mid \boldsymbol{x}) q(\boldsymbol{z}_i \mid \boldsymbol{z}_{pa(i)}) \log \left( \frac{q(c_{pa(j) \to j} \mid \boldsymbol{x})}{p(c_{pa(j) \to j} \mid \boldsymbol{z}_{pa(j)})} \right)
$$

$$
= \sum_{j \in \mathbb{V} \backslash \{0\}} \sum_{\mathcal{P}_{l \in j}} \int_{\boldsymbol{z}_{\mathcal{P}_l}} q(\boldsymbol{z}_0 \mid \boldsymbol{x}) \prod_{i \in \mathcal{P}_l \backslash \{0\}} q(c_{pa(i) \to i} \mid \boldsymbol{x}) q(\boldsymbol{z}_i \mid \boldsymbol{z}_{pa(i)}) \log \left( \frac{q(c_{pa(j) \to j} \mid \boldsymbol{x})}{p(c_{pa(j) \to j} \mid \boldsymbol{z}_{pa(j)})} \right) \tag{33}
$$

The above can be proved with the following Lemma, where we rewrite $\sum_{\mathcal{P}_{l \in j}} = \sum_{l \in \mathbb{L}} \mathbb{1}[j \in \mathcal{P}_l]$.

**Lemma A.1.** *Given a binary tree $\mathcal{T}$ as defined in Section 2.1, composed of a set of nodes $\mathbb{V} = \{0, \ldots, V\}$ and leaves $\mathbb{L} \subset \mathbb{V}$, where $\mathcal{P}_l$ is the decision path from root $0$ to leaf $l$, and $\boldsymbol{z}_{\mathcal{P}_l} = \{\boldsymbol{z}_i \mid i \in \mathcal{P}_l\}$ the set of latent variables selected by the path $\mathcal{P}_l$. Then it holds*

$$
\sum_{l \in \mathbb{L}} \int_{\boldsymbol{z}_{\mathcal{P}_l}} \sum_{j \in \mathcal{P}_l \backslash \{0\}} f(j, l, \boldsymbol{z}_{\mathcal{P}_l}) = \sum_{j \in \mathbb{V} \backslash \{0\}} \sum_{l \in \mathbb{L}} \int_{\boldsymbol{z}_{\mathcal{P}_l}} \mathbb{1}[j \in \mathcal{P}_l] f(j, l, \boldsymbol{z}_{\mathcal{P}_l}), \tag{34}
$$

*Proof.* The proof is as follows:

$$\sum_{j\in\mathbb{V}\backslash\{0\}}\sum_{l\in\mathbb{L}}\int_{\boldsymbol{z}_{\mathcal{P}_l}}\mathbb{1}[j\in\mathcal{P}_l]f(j,l,\boldsymbol{z}_{\mathcal{P}_l})=\sum_{j\in\mathbb{V}\backslash\{0\}}\sum_{l\in\mathbb{L}}\int_{\boldsymbol{z}_{\mathcal{P}_l}}f(j,l,\boldsymbol{z}_{\mathcal{P}_l})\sum_{i\in\mathcal{P}_l\backslash\{0\}}\mathbb{1}[i=j] \quad (35)$$

$$=\sum_{l\in\mathbb{L}}\sum_{j\in\mathbb{V}\backslash\{0\}}\int_{\boldsymbol{z}_{\mathcal{P}_l}}\sum_{i\in\mathcal{P}_l\backslash\{0\}}f(j,l,\boldsymbol{z}_{\mathcal{P}_l})\mathbb{1}[i=j] \quad (36)$$

$$=\sum_{l\in\mathbb{L}}\sum_{j\in\mathbb{V}\backslash\{0\}}\int_{\boldsymbol{z}_{\mathcal{P}_l}}\sum_{i\in\mathcal{P}_l\backslash\{0\}}f(i,l,\boldsymbol{z}_{\mathcal{P}_l})\mathbb{1}[i=j] \quad (37)$$

$$=\sum_{l\in\mathbb{L}}\int_{\boldsymbol{z}_{\mathcal{P}_l}}\sum_{i\in\mathcal{P}_l\backslash\{0\}}f(i,l,\boldsymbol{z}_{\mathcal{P}_l})\sum_{j\in\mathbb{V}\backslash\{0\}}\mathbb{1}[i=j] \quad (38)$$

$$=\sum_{l\in\mathbb{L}}\int_{\boldsymbol{z}_{\mathcal{P}_l}}\sum_{i\in\mathcal{P}_l\backslash\{0\}}f(i,l,\boldsymbol{z}_{\mathcal{P}_l}) \quad (39)$$

$$=\sum_{l\in\mathbb{L}}\int_{\boldsymbol{z}_{\mathcal{P}_l}}\sum_{j\in\mathcal{P}_l\backslash\{0\}}f(j,l,\boldsymbol{z}_{\mathcal{P}_l}). \quad (40)$$

$\square$

Having proven the equality, we can continue with the KL of the decisions as follows:

$$\mathrm{KL}_{decisions}=$$

$$\sum_{j\in\mathbb{V}\backslash\{0\}}\sum_{\mathcal{P}_{l\in j}}\int_{\boldsymbol{z}_{\mathcal{P}_l}}q(\boldsymbol{z}_0\mid\boldsymbol{x})\prod_{i\in\mathcal{P}_l\backslash\{0\}}q(c_{pa(i)\to i}\mid\boldsymbol{x})q(\boldsymbol{z}_i\mid\boldsymbol{z}_{pa(i)})\log\left(\frac{q(c_{pa(j)\to j}\mid\boldsymbol{x})}{p(c_{pa(j)\to j}\mid\boldsymbol{z}_{pa(j)})}\right) \quad (41)$$

$$=\sum_{j\in\mathbb{V}\backslash\{0\}}\sum_{\mathcal{P}_{l\in j}}\int_{\boldsymbol{z}_{\mathcal{P}_l}}[q(\boldsymbol{z}_0\mid\boldsymbol{x})\prod_{i\in\mathcal{P}_{\le j}\backslash\{0\}}q(c_{pa(i)\to i}\mid\boldsymbol{x})q(\boldsymbol{z}_i\mid\boldsymbol{z}_{pa(i)})\log\left(\frac{q(c_{pa(j)\to j}\mid\boldsymbol{x})}{p(c_{pa(j)\to j}\mid\boldsymbol{z}_{pa(j)})}\right)$$
$$\times\prod_{k\in\mathcal{P}_{>j}}q(c_{pa(k)\to k}\mid\boldsymbol{x})q(\boldsymbol{z}_k\mid\boldsymbol{z}_{pa(k)})] \quad (42)$$

$$=\sum_{j\in\mathbb{V}\backslash\{0\}}\sum_{\mathcal{P}_{>j}}\int_{\boldsymbol{z}_{\le j},\boldsymbol{z}_{>j}}[q(\boldsymbol{z}_0\mid\boldsymbol{x})\prod_{i\in\mathcal{P}_{\le j}\backslash\{0\}}q(c_{pa(i)\to i}\mid\boldsymbol{x})q(\boldsymbol{z}_i\mid\boldsymbol{z}_{pa(i)})\log\left(\frac{q(c_{pa(j)\to j}\mid\boldsymbol{x})}{p(c_{pa(j)\to j}\mid\boldsymbol{z}_{pa(j)})}\right)$$
$$\times\prod_{k\in\mathcal{P}_{>j}}q(c_{pa(k)\to k}\mid\boldsymbol{x})q(\boldsymbol{z}_k\mid\boldsymbol{z}_{pa(k)})] \quad (43)$$

$$=\sum_{j\in\mathbb{V}\backslash\{0\}}\int_{\boldsymbol{z}_{\le j}}q(\boldsymbol{z}_0\mid\boldsymbol{x})\prod_{i\in\mathcal{P}_{\le j}\backslash\{0\}}q(c_{pa(i)\to i}\mid\boldsymbol{x})q(\boldsymbol{z}_i\mid\boldsymbol{z}_{pa(i)})\log\left(\frac{q(c_{pa(j)\to j}\mid\boldsymbol{x})}{p(c_{pa(j)\to j}\mid\boldsymbol{z}_{pa(j)})}\right)$$
$$\times\sum_{\mathcal{P}_{>j}}\int_{\boldsymbol{z}_{>j}}\left[\prod_{k\in\mathcal{P}_{>j}}q(c_{pa(k)\to k}\mid\boldsymbol{x})q(\boldsymbol{z}_k\mid\boldsymbol{z}_{pa(k)})\right] \quad (44)$$

From Eq. 41 to Eq. 42, we split the inner product into the nodes of the paths $\mathcal{P}_{l\in j}$ that are before and after the node $j$.

From Eq. 42 to Eq. 43, we observe that the sum over all paths going through $j$ can be reduced to the sum over all paths starting from $j$, because there is only one path to $j$, which is specified in the product that comes after.

From Eq. 43 to Eq. 44, we observe that the sum over paths starting from $j$ and integral over $\boldsymbol{z}_{>j}$ concern only the terms of the second line. Observe that the term on the second line of Eq. 44

integrates out to 1 and we get

$$\mathrm{KL}_{decisions} =$$

$$\sum_{j \in \mathbb{V} \setminus \{0\}} \int_{\boldsymbol{z}_{\leq j}} q(\boldsymbol{z}_0 \mid \boldsymbol{x}) \prod_{i \in \mathcal{P}_{\leq j} \setminus \{0\}} q(c_{pa(i) \to i} \mid \boldsymbol{x}) q(\boldsymbol{z}_i \mid \boldsymbol{z}_{pa(i)}) \log \left( \frac{q(c_{pa(j) \to j} \mid \boldsymbol{x})}{p(c_{pa(j) \to j} \mid \boldsymbol{z}_{pa(j)})} \right) \quad (45)$$

$$= \sum_{j \in \mathbb{V} \setminus \{0\}} \int_{\boldsymbol{z}_{< j}} \int_{\boldsymbol{z}_j} q(\boldsymbol{z}_0 \mid \boldsymbol{x}) \prod_{i \in \mathcal{P}_{\leq j} \setminus \{0\}} q(c_{pa(i) \to i} \mid \boldsymbol{x}) q(\boldsymbol{z}_i \mid \boldsymbol{z}_{pa(i)}) \log \left( \frac{q(c_{pa(j) \to j} \mid \boldsymbol{x})}{p(c_{pa(j) \to j} \mid \boldsymbol{z}_{pa(j)})} \right) \quad (46)$$

$$= \sum_{j \in \mathbb{V} \setminus \{0\}} \int_{\boldsymbol{z}_{< j}} q(\boldsymbol{z}_0 \mid \boldsymbol{x}) \prod_{i \in \mathcal{P}_{< j} \setminus \{0\}} q(c_{pa(i) \to i} \mid \boldsymbol{x}) q(\boldsymbol{z}_i \mid \boldsymbol{z}_{pa(i)}) \log \left( \frac{q(c_{pa(j) \to j} \mid \boldsymbol{x})}{p(c_{pa(j) \to j} \mid \boldsymbol{z}_{pa(j)})} \right)$$
$$\times \int_{\boldsymbol{z}_j} q(c_{pa(j) \to j} \mid \boldsymbol{x}) q(\boldsymbol{z}_j \mid \boldsymbol{z}_{pa(j)}) \quad (47)$$

$$= \sum_{j \in \mathbb{V} \setminus \{0\}} \int_{\boldsymbol{z}_{< j}} q(\boldsymbol{z}_0 \mid \boldsymbol{x}) \prod_{i \in \mathcal{P}_{< j} \setminus \{0\}} q(c_{pa(i) \to i} \mid \boldsymbol{x}) q(\boldsymbol{z}_i \mid \boldsymbol{z}_{pa(i)}) q(c_{pa(j) \to j} \mid \boldsymbol{x})$$
$$\times \log \left( \frac{q(c_{pa(j) \to j} \mid \boldsymbol{x})}{p(c_{pa(j) \to j} \mid \boldsymbol{z}_{pa(j)})} \right). \quad (48)$$

From Eq. 46 to Eq. 47, we single out the term in the product that corresponds to $j = i$, which is the only term that depends on $\int_{z_i}$.

From Eq. 47 to Eq. 48, we observe that in the singled-out term, $\int_{z_j} q(z_j \mid \boldsymbol{z}_{pa(j)}) = 1$, which leaves only $q(c_{pa(j) \to j} \mid \boldsymbol{x})$.

This equation can be rewritten in a more interpretable way. Let us define the probability of reaching node $j$ and observing $\boldsymbol{z}_j$ as

$$P(j; \boldsymbol{z}, \boldsymbol{c}) = q(\boldsymbol{z}_0 \mid \boldsymbol{x}) \prod_{i \in \mathcal{P}_{\leq j} \setminus \{0\}} q(c_{pa(i) \to i} \mid \boldsymbol{x}) q(\boldsymbol{z}_i \mid \boldsymbol{z}_{pa(i)}). \quad (49)$$

Then the KL term of the decisions can be simplified as

$$\mathrm{KL}_{decisions} = \sum_{j \in \mathbb{V} \setminus \{0\}} \int_{\boldsymbol{z}_{< j}} P(pa(j); \boldsymbol{z}, \boldsymbol{c}) q(c_{pa(j) \to j} \mid \boldsymbol{x}) \log \left( \frac{q(c_{pa(j) \to j} \mid \boldsymbol{x})}{p(c_{pa(j) \to j} \mid \boldsymbol{z}_{pa(j)})} \right) \quad (50)$$

$$= \sum_{i \in \mathbb{V} \setminus \mathbb{L}} \sum_{k \in \{0,1\}} \int_{\boldsymbol{z}_{< i}} P(i; \boldsymbol{z}, \boldsymbol{c}) q(c_i = k \mid \boldsymbol{x}) \log \left( \frac{q(c_i = k \mid \boldsymbol{x})}{p(c_i = k \mid \boldsymbol{z}_i)} \right). \quad (51)$$

This term requires Monte Carlo sampling for the expectations over the latent variables $\boldsymbol{z}$, while we can analytically compute the sum over all decisions $\mathcal{P}_l$.

$$\mathrm{KL}_{decisions} = \sum_{i \in \mathbb{V} \setminus \mathbb{L}} \int_{\boldsymbol{z}_{< i}} P(i; \boldsymbol{z}, \boldsymbol{c})$$
$$\times \left[ q(\mathsf{c}_i = 0) \mid \boldsymbol{x}) \log \left( \frac{q(\mathsf{c}_i = 0 \mid \boldsymbol{x})}{p(\mathsf{c}_i = 0 \mid \boldsymbol{z}_i)} \right) + q(\mathsf{c}_i = 1 \mid \boldsymbol{x}) \log \left( \frac{q(\mathsf{c}_i = 1 \mid \boldsymbol{x})}{p(\mathsf{c}_i = 1 \mid \boldsymbol{z}_i)} \right) \right] \quad (52)$$

$$\approx \frac{1}{M} \sum_{m=1}^{M} \sum_{i \in \mathbb{V} \setminus \mathbb{L}} P(i; \boldsymbol{z}^{(m)}, \boldsymbol{c})$$
$$\times \left[ q(\mathsf{c}_i = 0) \mid \boldsymbol{x}) \log \left( \frac{q(\mathsf{c}_i = 0 \mid \boldsymbol{x})}{p(\mathsf{c}_i = 0 \mid \boldsymbol{z}_i^{(m)})} \right) + q(\mathsf{c}_i = 1 \mid \boldsymbol{x}) \log \left( \frac{q(\mathsf{c}_i = 1 \mid \boldsymbol{x})}{p(\mathsf{c}_i = 1 \mid \boldsymbol{z}_i^{(m)})} \right) \right], \quad (53)$$

where $P(i; \boldsymbol{z}^{(m)}, \boldsymbol{c}) = \prod_{j \in \mathcal{P}_{\leq i}} q(c_{pa(j) \to j} \mid \boldsymbol{x})$ is the probability of reaching node $i$, defined as $P(i; \boldsymbol{c})$ in Eq. 16/19/20 for simplicity.

### A.1.3 KL Nodes

Finally, we can analyze the last term of the KL term, which corresponds to the KL of the nodes (26). The reasoning is similar to the equations above and we will use the same notation. The KL of the nodes can be written as

$$
\mathrm{KL}_{nodes} = \sum_{l \in \mathbb{L}} \int_{\boldsymbol{z}_{\mathcal{P}_l}} q(\boldsymbol{z}_0 \mid \boldsymbol{x}) \prod_{i \in \mathcal{P}_l \setminus \{0\}} q(c_{pa(i) \to i} \mid \boldsymbol{x}) q(\boldsymbol{z}_i \mid \boldsymbol{z}_{pa(i)})
$$
$$
\times \log \left( \frac{\prod_{j \in \mathcal{P}_l \setminus \{0\}} q(\boldsymbol{z}_j \mid \boldsymbol{z}_{pa(j)})}{\prod_{k \in \mathcal{P}_l \setminus \{0\}} p(\boldsymbol{z}_k \mid \boldsymbol{z}_{pa(k)})} \right) \tag{54}
$$
$$
= \sum_{l \in \mathbb{L}} \int_{\boldsymbol{z}_{\mathcal{P}_l}} \sum_{j \in \mathcal{P}_l \setminus \{0\}} q(\boldsymbol{z}_0 \mid \boldsymbol{x}) \prod_{i \in \mathcal{P}_l \setminus \{0\}} q(c_{pa(i) \to i} \mid \boldsymbol{x}) q(\boldsymbol{z}_i \mid \boldsymbol{z}_{pa(i)}) \log \left( \frac{q(\boldsymbol{z}_j \mid \boldsymbol{z}_{pa(j)})}{p(\boldsymbol{z}_j \mid \boldsymbol{z}_{pa(j)})} \right) \tag{55}
$$

We now change from a pathwise view to a nodewise view.

$$
= \sum_{j \in \mathbb{V} \setminus \{0\}} \sum_{\mathcal{P}_{l \ni j}} \int_{\boldsymbol{z}_{\leq j}, \boldsymbol{z}_{> j}} q(\boldsymbol{z}_0 \mid \boldsymbol{x}) \prod_{i \in \mathcal{P}_l \setminus \{0\}} q(c_{pa(i) \to i} \mid \boldsymbol{x}) q(\boldsymbol{z}_i \mid \boldsymbol{z}_{pa(i)}) \log \left( \frac{q(\boldsymbol{z}_j \mid \boldsymbol{z}_{pa(j)})}{p(\boldsymbol{z}_j \mid \boldsymbol{z}_{pa(j)})} \right) \tag{56}
$$
$$
= \sum_{j \in \mathbb{V} \setminus \{0\}} \int_{\boldsymbol{z}_{\leq j}} q(\boldsymbol{z}_0 \mid \boldsymbol{x}) \prod_{i \in \mathcal{P}_{\leq j} \setminus \{0\}} q(c_{pa(i) \to i} \mid \boldsymbol{x}) q(\boldsymbol{z}_i \mid \boldsymbol{z}_{pa(i)}) \log \left( \frac{q(\boldsymbol{z}_j \mid \boldsymbol{z}_{pa(j)})}{p(\boldsymbol{z}_j \mid \boldsymbol{z}_{pa(j)})} \right)
$$
$$
\times \sum_{\mathcal{P}_{> j}} \int_{\boldsymbol{z}_{> j}} \left[ \prod_{k \in \mathcal{P}_{> j}} q(c_{pa(k) \to k} \mid \boldsymbol{x}) q(\boldsymbol{z}_k \mid \boldsymbol{z}_{pa(k)}) \right] \tag{57}
$$
$$
= \sum_{j \in \mathbb{V} \setminus \{0\}} \int_{\boldsymbol{z}_{< j}} \int_{\boldsymbol{z}_j} q(\boldsymbol{z}_0 \mid \boldsymbol{x}) \prod_{i \in \mathcal{P}_{\leq j} \setminus \{0\}} q(c_{pa(i) \to i} \mid \boldsymbol{x}) q(\boldsymbol{z}_i \mid \boldsymbol{z}_{pa(i)}) \log \left( \frac{q(\boldsymbol{z}_j \mid \boldsymbol{z}_{pa(j)})}{p(\boldsymbol{z}_j \mid \boldsymbol{z}_{pa(j)})} \right) \tag{58}
$$
$$
= \sum_{j \in \mathbb{V} \setminus \{0\}} \int_{\boldsymbol{z}_{< j}} \int_{\boldsymbol{z}_j} P(pa(j); \boldsymbol{z}, \boldsymbol{c}) q(c_{pa(j) \to j} \mid \boldsymbol{x}) q(\boldsymbol{z}_j \mid \boldsymbol{z}_{pa(j)}) \log \left( \frac{q(\boldsymbol{z}_j \mid \boldsymbol{z}_{pa(j)})}{p(\boldsymbol{z}_j \mid \boldsymbol{z}_{pa(j)})} \right) \tag{59}
$$
$$
= \sum_{j \in \mathbb{V} \setminus \{0\}} \int_{\boldsymbol{z}_{< j}} P(pa(j); \boldsymbol{z}, \boldsymbol{c}) q(c_{pa(j) \to j} \mid \boldsymbol{x}) \int_{\boldsymbol{z}_j} q(\boldsymbol{z}_j \mid \boldsymbol{z}_{pa(j)}) \log \left( \frac{q(\boldsymbol{z}_j \mid \boldsymbol{z}_{pa(j)})}{p(\boldsymbol{z}_j \mid \boldsymbol{z}_{pa(j)})} \right) \tag{60}
$$
$$
= \sum_{j \in \mathbb{V} \setminus \{0\}} \int_{\boldsymbol{z}_{< j}} P(pa(j); \boldsymbol{z}, \boldsymbol{c}) q(c_{pa(j) \to j} \mid \boldsymbol{x}) \mathrm{KL} \left( q(\boldsymbol{z}_j \mid \boldsymbol{z}_{pa(j)}) \mid p(\boldsymbol{z}_j \mid \boldsymbol{z}_{pa(j)}) \right) \tag{61}
$$
$$
\approx \frac{1}{M} \sum_{m=1}^{M} \sum_{i \in \mathbb{V} \setminus \{0\}} P(pa(i); \boldsymbol{z}^{(m)}, \boldsymbol{c}) q(c_{pa(i) \to i} \mid \boldsymbol{x})
$$
$$
\times \mathrm{KL}(q(\boldsymbol{z}_i^{(m)} \mid pa(\boldsymbol{z}_i^{(m)})) \| p(\boldsymbol{z}_i^{(m)} \mid pa(\boldsymbol{z}_i^{(m)}))) \tag{62}
$$
$$
= \frac{1}{M} \sum_{m=1}^{M} \sum_{i \in \mathbb{V} \setminus \{0\}} P(i; \boldsymbol{z}^{(m)}, \boldsymbol{c}) \mathrm{KL}(q(\boldsymbol{z}_i^{(m)} \mid pa(\boldsymbol{z}_i^{(m)})) \| p(\boldsymbol{z}_i^{(m)} \mid pa(\boldsymbol{z}_i^{(m)}))), \tag{63}
$$

where $P(pa(j); \boldsymbol{z}, \boldsymbol{c})$ is defined in Eq. 49 and where $P(i; \boldsymbol{z}^{(m)}, \boldsymbol{c}) = P(i; \boldsymbol{c}) = \prod_{j \in \mathcal{P}_{\leq i}} q(c_{pa(j) \to j} \mid \boldsymbol{x})$ is the probability of reaching node $i$.

### A.1.4 KL terms

Using the above factorization, the KL term of the ELBO can be written as

$$
\begin{aligned}
\text{KL}\left(q\left(\boldsymbol{z}, \mathcal{P}_l \mid \boldsymbol{x}\right) \| p\left(\boldsymbol{z}, \mathcal{P}_l\right)\right) \approx {}& \text{KL}\left(q(\boldsymbol{z}_0 \mid \boldsymbol{x}) \| p(\boldsymbol{z}_0)\right) \\
& + \frac{1}{M} \sum_{m=1}^{M} \sum_{i \in \mathbb{V} \setminus \mathbb{L}} P(i; \boldsymbol{z}^{(m)}, \boldsymbol{c}) \sum_{c_i \in \{0,1\}} q(c_i \mid \boldsymbol{x}) \log\left(\frac{q(c_i \mid \boldsymbol{x})}{p(c_i \mid \boldsymbol{z}_i^{(m)}))}\right) \\
& + \frac{1}{M} \sum_{m=1}^{M} \sum_{i \in \mathbb{V} \setminus \{0\}} P(i; \boldsymbol{z}^{(m)}, \boldsymbol{c}) \, \text{KL}(q(\boldsymbol{z}_i^{(m)} \mid pa(\boldsymbol{z}_i^{(m)})) \| p(\boldsymbol{z}_i^{(m)} \mid pa(\boldsymbol{z}_i^{(m)}))),
\end{aligned}
\tag{64}
$$

where $P(i; \boldsymbol{z}^{(m)}, \boldsymbol{c}) = P(i; \boldsymbol{c}) = \prod_{j \in \mathcal{P}_{\leq i}} q(c_{pa(j) \to j} \mid \boldsymbol{x})$.

### A.1.5 Reconstruction Loss

Finally, to compute the full ELBO, the KL terms are added to the reconstruction loss defined in (16). Here, assumptions about the distribution of the inputs are required. For the grayscale datasets such as MNIST, Fashion-MNIST, and Omniglot, as well as the one-hot-encoded 20Newsgroup, we assume that the inputs are Bernoulli distributed, such that the resulting reconstruction loss is the binary cross entropy. On the other hand, for the colored datasets CIFAR-10, CIFAR-100, and CelebA, we assume that their pixel values are normally distributed, which leads to the mean squared error as loss function, where we assume that $\sigma = 1$.

## A.2 Computational Complexity

All terms of the Evidence Lower Bound, Equation 12, can be computed efficiently and the computational complexity of a single joint update of the parameters is $O(MBVDC_p)$, where $M$ is the number of MC samples, $B$ is the batch size, $V$ is the number of nodes in the tree, $D$ is the maximum depth of the tree, and $C_p$ is the cost to compute the KL between Gaussians. It should be noted that the computational complexity is, in practice, reduced to $O(LBVC_p)$, as the term $P(i; \boldsymbol{c})$ can be computed dynamically from parent nodes.

# B Related Work

In addition to the review presented in Section 3, which encompasses a broad range of related work in the field of deep latent variable models, we provide a review of relevant work in the domains of hierarchical clustering and decision trees. By doing so, we hope to shed further light on the current state-of-the-art approaches and contribute to a deeper understanding of the challenges and opportunities that lie in the intersection of hierarchical clustering, decision trees, and latent variable models.

## B.1 Hierarchical Clustering

Hierarchical clustering algorithms have long been employed in the field of data mining and machine learning to extract hierarchical structures from data (Sneath, 1957; Ward, 1963; Murtagh & Contreras, 2012). Agglomerative clustering is among the earliest and most well-known hierarchical clustering algorithms. These methods start with each data point as an individual cluster and then iteratively merge the closest pairs of clusters, according to a predefined distance metric, until a stopping criterion is met. While single-linkage and complete-linkage agglomeration clustering are widely used as baselines, we observe better performance when using the bottom-up strategy proposed by Ward (1963). Ward's minimum variance criterion minimizes the total within-cluster variance (Murtagh & Legendre, 2014), thus providing balanced and compact clusters. In contrast, the Bayesian Hierarchical Clustering (BHC) proposed by Heller & Ghahramani (2005) takes a different approach by employing hypothesis testing to determine when to merge the clusters. The divisive clustering algorithms, on the other hand, provide a different strategy to hierarchical clustering. Unlike agglomerative methods, divisive clustering starts with all data points in a single cluster and recursively splits clusters into smaller ones. The proposed TreeVAE is an example of a divisive clustering method. Among a variety of proposed methods, the Bisecting-K-means algorithm (Steinbach et al., 2000; Nistér & Stewénius,

2006) is widely used for its simplicity; it applies k-means with two clusters recursively. Similarly, Williams (1999) learn a hierarchical probabilistic Gaussian mixture model. Neal (2003) introduce a hierarchical generalization of Dirichlet process mixture models.

More recent approaches include PERCH (Kobren et al., 2017b), which is a non-greedy, incremental algorithm that scales to both the number of data points and the number of clusters, GHC (Monath et al., 2019), which leverages continuous representations of trees in a hyperbolic space and optimizes a differentiable cost function, and Mathieu et al. (2019), which similarly use a Poincaré ball model of hyperbolic geometry as a latent space to learn continuous hierarchical representations with a VAE.Furthermore, RSSCOMP (You et al., 2015), which explores a subspace clustering method based on an orthogonal matching pursuit. Finally, Deep ECT (Mautz et al., 2020) proposes a divisive hierarchical embedded clustering method, which jointly optimizes an autoencoder that compresses the data into an embedded space and a hierarchical clustering layer on top of it. Density-based clustering algorithms, such as DBSCAN, belong to a distinct category of clustering techniques. They aim to identify regions in a dataset where points are densely concentrated and classify outliers as noise (Ester et al., 1996). R. J. Campello et al. (2015) build on this idea to learn a hierarchy based on the distances between datapoints where distance is roughly determined by the density. However, one limitation of density-based methods lie in their performance when confronted with complex datasets requiring high-dimensional representations. In such cases, estimating density requires an exponentially growing number of data points, which leads to scalability issues that TreeVAE does not have.

## B.2   Decision Trees

Decision trees (Breiman et al., 1984) are interpretable, non-parametric supervised learning techniques commonly employed in classification and regression tasks. They rely on the data itself to build a hierarchical structure. This is done by recursively partitioning the data into subsets, each of which corresponds to a specific node in the tree. At each node, a decision rule is generated based on one of the input features that best discriminates the data in that subset. One of the key advantages of decision trees is their interpretability. The learned tree structure can be easily visualized to get insights into the data and the model's decision-making process. Suárez & Lutsko (1999) argue that deterministic splits lead to overfitting and introduce fuzzy decision trees with probabilistic decisions, implicitly allowing for backpropagation. With the advancement of neural networks, many works (Rota Bulo & Kontschieder, 2014; Laptev & Buhmann, 2014; Frosst & Hinton, 2017) leverage MLPs or CNNs for learning a more complex decision rule. However, the input at every node remains the original features, which limits their performance, as they are unable to learn meaningful representations. Thus, Tanno et al. (2019) introduces Adaptive Neural Trees (ANT), a method that learns flexible, hierarchical representations through NNs, hereby facilitating hierarchical separation of task-relevant features. Additionally, ANTs architectures grow dynamically such that they can adapt to the complexity of the training dataset. At inference time, ANTs allow for lightweight conditional computation via the most likely path, leading to inbuilt interpretable decision-making.

While decision trees were initially designed with the goal of achieving high predictive performance, they are also used for many auxiliary goals such as semi-supervised learning (Zharmagambetov & Carreira-Perpiñán, 2022), robustness (Moshkovitz et al., 2021) or interpretability (Wan et al., 2021; Souza et al., 2022; Arenas et al., 2022; Pace et al., 2022). In the context of interpretability, various approaches have been proposed to enhance accuracy and interpretability. Wan et al. (2021) introduce Neural-Backed Decision Trees (NBDTs), which improve accuracy by replacing the final layer of a neural network with a differentiable sequence of decisions to increase its interpretability while retaining predictive performance. Souza et al. (2022) focus on optimizing the structural parameters of decision trees, introducing the concept of "explanation size" as being the expected number of attributes required for prediction to measure interpretability. Pace et al. (2022) develop the POETREE framework for interpretable policy learning via decision trees in time-varying clinical decision environments. Arenas et al. (2022) investigate explanations in decision trees, considering both deterministic and probabilistic approaches and showing the limitations thereof. These works collectively contribute to advancing the use of decision trees in interpretable machine learning, providing insights into trade-offs, criteria, and frameworks for improving accuracy and interpretability.

While decision trees are most commonly known for their application in supervised tasks, they can be repurposed to partition the data space into clusters in an unsupervised way (Blockeel & De Raedt, 1998; B. Liu et al., 2000; Basak & Krishnapuram, 2005; Ram & Gray, 2011; Fraiman et al., 2013). Most methods, however, have the downside of not learning meaningful representations for their splits, such that they are unfit for modelling complex interactions. Therefore, similar to Tanno et al. (2019) in the supervised setting, in this work, we combine the simplicity and interpretability of clustering decision trees with the flexibility of NNs.

## C    Further Experiments

Here, we provide additional experimental results, including further hierarchical structures, unconditional generation of samples, and ablation experiments.

### C.1    Discovery of Hierarchies

We show further hierarchical structures learned by TreeVAE on MNIST in Fig. 9/10, Fashion-MNIST in Fig. 11, Omniglot-50 in Fig. 12, and CIFAR-100 in Fig. 13. Additionally, the figures that include conditional generations are generated by sampling from the root and following the most probable path to generate every leaf-specific image.

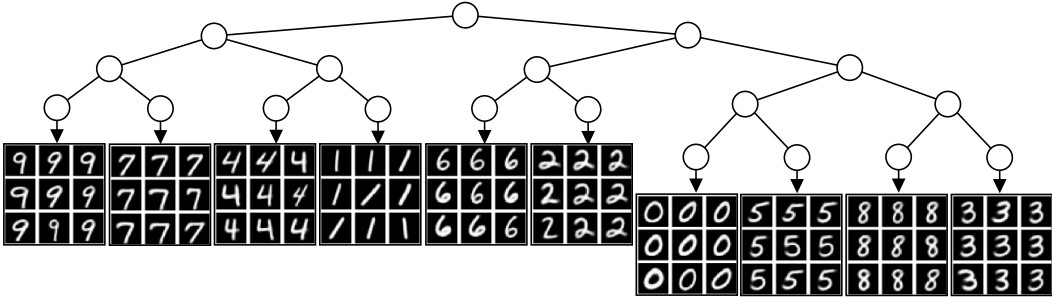

Figure 9: Hierarchical structure learned by TreeVAE with ten leaves on MNIST dataset with generated images through conditional sampling. The right subtree contains rounded digits, while the left subtree contains digits with a straight line.

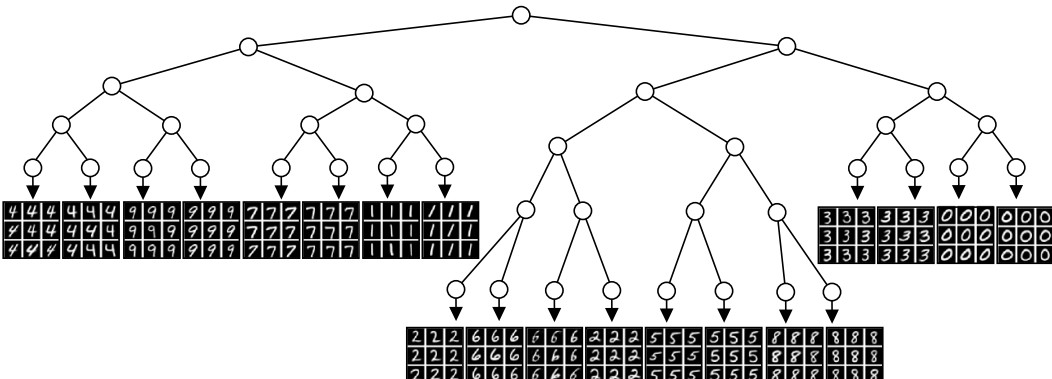

Figure 10: Hierarchical structure learned by TreeVAE with 20 leaves on MNIST dataset with generated images through conditional sampling. The digits are further divided according to style.

### C.2    Generations

We further show how TreeVAE can be used to sample unconditional generations for all clusters simultaneously for MNIST, in Fig. 14, and CelebA, in Fig. 15. Differently from the conditional generation, where we follow the path of the tree with the highest probability, here we follow all

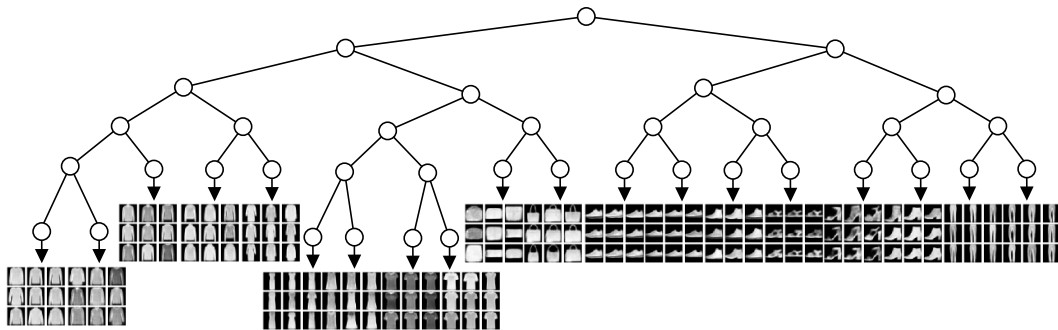

Figure 11: Hierarchical structure learned by TreeVAE with 20 leaves on Fashion-MNIST dataset with generated images.

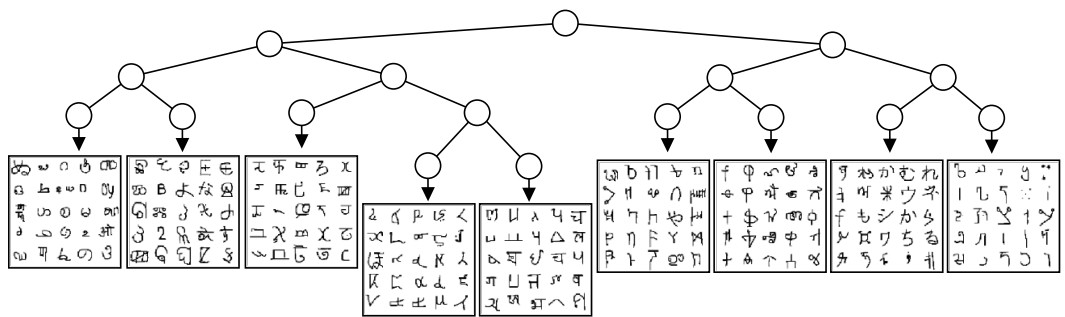

Figure 12: Hierarchical structure learned by TreeVAE on the full Omniglot dataset. We display random subsets of images that are probabilistically assigned to each leaf of the tree. Similar to the results in the main text, we can again find regional hierarchies in some individual subtrees. For instance, the leftmost subtree seems to find structures in different Indian alphabets. We can also observe that this subtree seems to cluster smaller characters in its left child, whereas the right child contains bigger shapes. Another example is the rightmost tree, which seems to encode the more straight shapes of Japanese writing styles. There, the left child encodes more complex shapes that contain many different strokes, and the right subtree groups simpler shapes with only a few lines.

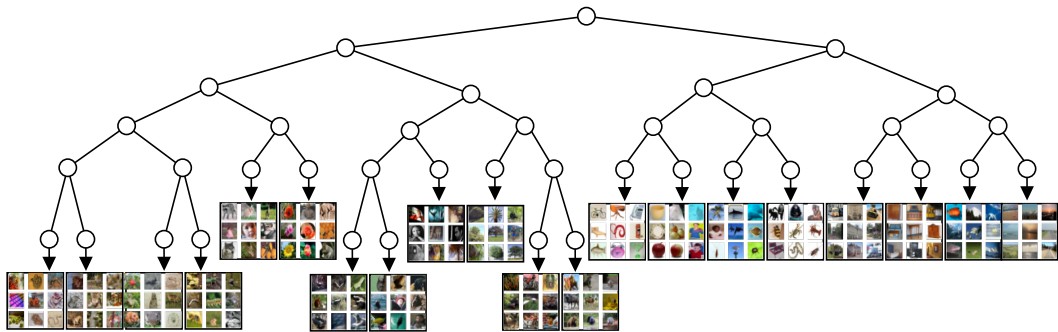

Figure 13: Hierarchical structure learned by TreeVAE with 20 leaves on CIFAR-100 dataset. We display random subsets of images that are probabilistically assigned to each leaf of the tree.

paths of the tree regardless of their probabilities. As a result, we generate $L$ images, where $L$ is the number of leaves, corresponding to the number of decoders. These generated samples exhibit distinct characteristics based on their respective cluster-specific features while maintaining cluster-independent properties across all generated instances.

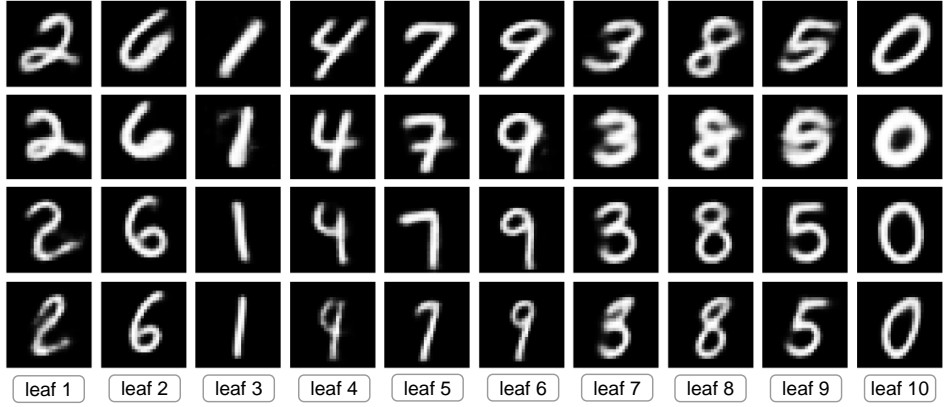

Figure 14: Selected unconditional generations of MNIST. One row corresponds to one sample from the root, for which we depict the visualizations obtained from the 10 leaf-decoders. Each row retains similar properties across the different leaves. In the first row, all digits are rotated; the second are bold; the third are straight; and the last are squeezed.

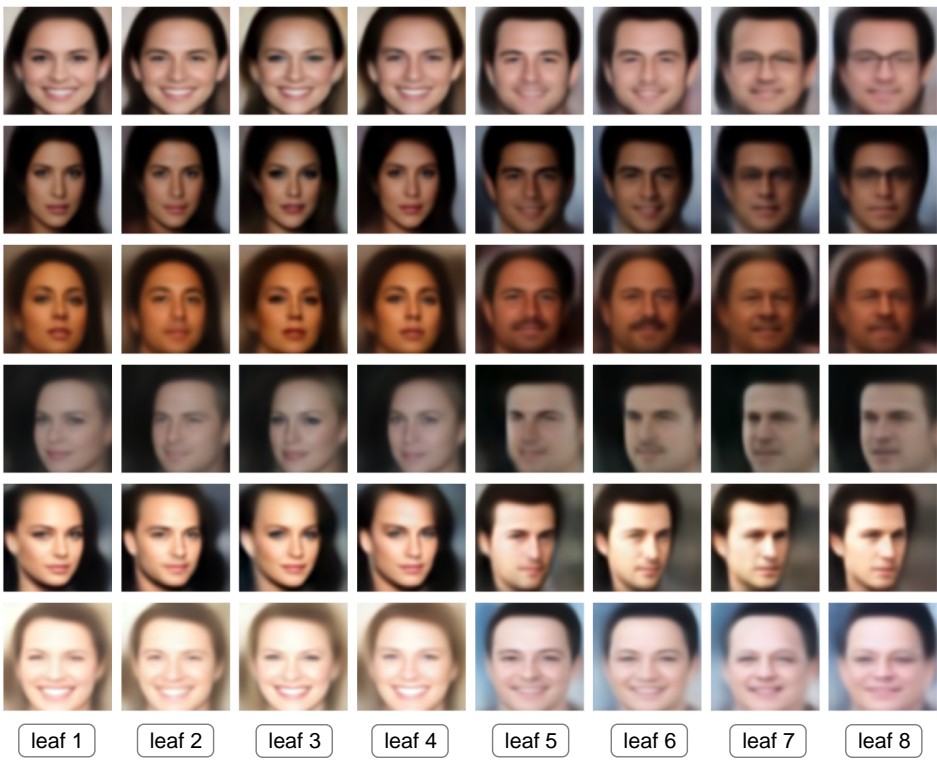

Figure 15: Selected unconditional generations of CelebA. One row corresponds to one sample from the root, for which we depict the visualizations obtained from the 8 leaf-decoders. The overall face shape, skin color, and face orientation are retained among leaves from the same row, while several properties (such as beard, mustache, hair) vary across the different leaves.

## C.3 Ablation Study

In this section, we provide additional studies on the effect of selected hyperparameters and assumptions on the behavior of TreeVAE. In Table 4, we provide an ablation on selected hyperparameters of TreeVAE on Fashion MNIST. In the following subsection we will provide further in-depth explorations of TreeVAE.

Table 4: Ablation Study on Fashion MNIST: (A) dimensionality of latent embeddings ordered by depth, (B) dimensionality of MLP and Router hidden layer ordered by depth, (C) Increasing dimensionality, (D) number of hidden layers of MLP, (E) number of hidden layers of router, (F) pre-determined maximum depth of tree.

| | $\mathbf{z}_{depth}$ | $MLP_{depth}$ | $MLP_{layers}$ | $Router_{layers}$ | $depth$ | Acc | NMI | ELBO |
|---|---|---|---|---|---|---|---|---|
| base | $8,8,8,8,8,8$ | $128,128,128,128,128$ | 1 | 2 | 6 | $63.5\pm4.1$ | $63.9\pm2.2$ | $-239.2\pm0.3$ |
| (A) | $2,2,2,2,2,2$ | | | | | $49.8\pm4.0$ | $51.8\pm2.7$ | $-249.5\pm0.4$ |
| (A) | $4,4,4,4,4,4$ | | | | | $60.0\pm4.7$ | $61.0\pm3.7$ | $-242.6\pm0.4$ |
| (A) | $16,16,16,16,16,16$ | | | | | $61.0\pm4.5$ | $63.0\pm2.4$ | $-238.8\pm0.4$ |
| (A) | $32,32,32,32,32,32$ | | | | | $62.7\pm4.4$ | $63.5\pm2.8$ | $-238.7\pm0.4$ |
| (B) | | $16,16,16,16,16$ | | | | $58.3\pm3.5$ | $60.6\pm2.1$ | $-241.5\pm0.4$ |
| (B) | | $32,32,32,32,32$ | | | | $60.6\pm3.6$ | $62.7\pm2.5$ | $-240.4\pm0.4$ |
| (B) | | $64,64,64,64,64$ | | | | $62.6\pm3.8$ | $63.5\pm1.4$ | $-239.6\pm0.3$ |
| (B) | | $256,256,256,256,256$ | | | | $62.3\pm4.9$ | $63.4\pm3.9$ | $-239.2\pm0.4$ |
| (C) | $64,32,16,8,4,2$ | $128,64,32,16,8$ | | | | $56.7\pm4.0$ | $59.9\pm3.1$ | $-239.2\pm0.4$ |
| (C) | $64,32,16,8,4,2$ | $256,128,64,32,16$ | | | | $60.9\pm5.3$ | $62.6\pm2.4$ | $-238.6\pm0.3$ |
| (C) | $128,64,32,16,8,4$ | $128,64,32,16,8$ | | | | $58.5\pm4.4$ | $61.3\pm2.5$ | $-239.1\pm0.4$ |
| (C) | $128,64,32,16,8,4$ | $256,128,64,32,16$ | | | | $62.6\pm3.4$ | $63.6\pm1.5$ | $-238.8\pm0.3$ |
| (C) | $256,128,64,32,16,8$ | $256,128,64,32,16$ | | | | $61.1\pm3.7$ | $61.7\pm2.9$ | $-238.9\pm0.3$ |
| (C) | $256,128,64,32,16,8$ | $512,256,128,64,32$ | | | | $60.5\pm4.7$ | $61.1\pm3.1$ | $-239.1\pm0.3$ |
| (D) | | | 2 | | | $60.3\pm5.3$ | $61.6\pm3.5$ | $-239.0\pm0.3$ |
| (D) | | | 3 | | | $61.4\pm4.2$ | $62.9\pm3.2$ | $-239.0\pm0.3$ |
| (E) | | | | 1 | | $59.6\pm4.0$ | $62.3\pm2.0$ | $-239.2\pm0.4$ |
| (E) | | | | 3 | | $62.1\pm4.4$ | $63.6\pm3.1$ | $-239.1\pm0.4$ |
| (F) | $8,8,8,8,8$ | $128,128,128,128$ | | | 5 | $64.0\pm4.6$ | $64.4\pm2.7$ | $-239.2\pm0.3$ |
| (F) | $8,8,8,8,8,8,8$ | $128,128,128,128,128,128$ | | | 7 | $61.0\pm4.9$ | $62.9\pm2.9$ | $-239.2\pm0.3$ |
| (F) | $8,8,8,8,8,8,8,8$ | $128,128,128,128,128,128,128$ | | | 8 | $62.1\pm5.2$ | $63.6\pm2.7$ | $-239.1\pm0.4$ |

### C.3.1 Training time parameter matching

TreeVAE has the advantage of lightweight inference, as each datum ends up in one leaf. Still, we have to learn the full tree with all paths during training. Thus, we have conducted an ablation study on Fashion MNIST that increases the effective number of parameters of LadderVAE to match the parameter count of TreeVAE at training instead of inference. The results are presented in Table 5.

Table 5: Test set performances (%) of TreeVAE and LadderVAE with matched training parameter count. Means and standard deviations are computed across 10 runs with different random model initializations.

| Dataset | Method | ACC | NMI | LL | RL | ELBO |
|---|---|---|---|---|---|---|
| Fashion | LadderVAE + Agg | $56.4\pm1.9$ | $62.3\pm2.0$ | $-233.7\pm0.1$ | $224.8\pm0.3$ | $-238.9\pm0.3$ |
| | TreeVAE | $63.6\pm3.3$ | $64.7\pm1.4$ | $-234.7\pm0.1$ | $226.5\pm0.3$ | $-239.2\pm0.4$ |

### C.3.2 Tree Structure

First, we analyze the necessity of the iterative growing procedure as opposed to fixing an (informed) tree structure a priori. Thus, we present an ablation, where we fix a reasonable tree structure and directly train the full tree as opposed to iteratively grow. The results in Table 6 suggest that the iterative learning is vital to the performance of TreeVAE. Intuitively, by fixing a tree structure, the model loses the ability to learn one out of multiple equivalent tree structures (e.g. left and right subtrees can be exchanged) and instead needs to fit one specific structure. This leads to less flexibility and more influence of the random weight initialization, as well as local optima.

Second, we analyze whether the learnt tree structure is informed through the learnt embeddings or rather an artifact of induced biases. For this, we show the UMAP (Ghojogh et al., 2021) representation of the learnt root embedding in Fig. 16. As can be seen, tops as well as shoes are clustered together, while bags are somewhat in between. Trousers on the other hand are completely separate. Our interpretation is that the root split is made between shoes and tops, where it is unclear in which subtree bags and trousers should fall as they are both not clearly assigned to one of the two groups. Therefore, depending on initialization, they might end up on either side. To analyze the effect of

Table 6: Test set performances (%) of TreeVAE with a fixed structure standard TreeVAE. Means and standard deviations are computed across 10 runs with different random model initializations.

| Dataset | Method | ACC | NMI | LL | RL | ELBO |
|---------|--------|-----|-----|----|----|------|
| Fashion | fixed TreeVAE | $35.1 \pm 4.8$ | $54.1 \pm 5.8$ | $-237.4 \pm 0.4$ | $228.9 \pm 0.5$ | $-241.0 \pm 0.5$ |
|         | TreeVAE | $63.6 \pm 3.3$ | $64.7 \pm 1.4$ | $-234.7 \pm 0.1$ | $226.5 \pm 0.3$ | $-239.2 \pm 0.4$ |

random weight initialization on the learned dendrogram, we have made further efforts to find a dendrogram that best summarizes the learned dendrogram of multiple runs. We show it in Fig. 17. For this, we first align the leaves of the different trees by maximizing the overlapping samples, then we store the number of edges between any two clusters for every tree and then average this number over all trees. Thus, we have computed a distance matrix of all clusters, which is averaged over all trees. We then used average and complete linkage to cluster according to the average distance matrix. The recovered dendrograms of the two algorithms are identical, apart from the Bag and the Trouser cluster, which switch places. That is, all tops are in one subtree and all shoes in the other, while bags and trousers are assigned to either the shoes or tops subtree, depending on the clustering algorithm used. What this indicates is that the groups of shoes and tops are consistently recovered, no matter the random initialization, while the assignment of bags and trousers varies, which is aligned with the interpretation of the root embedding in Fig. 16, as there is no clear assignment thereof.

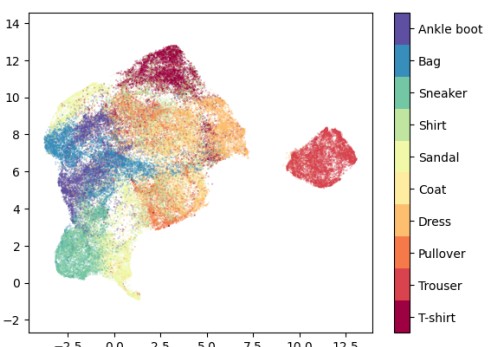

Figure 16: UMAP visualization of root embedding for Fashion MNIST.

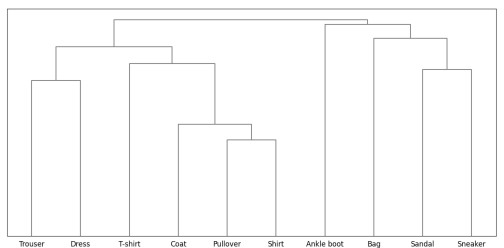

Figure 17: "Average" dendrogram learnt on Fashion MNIST by averaging cluster distances across 10 runs and applying average linkage clustering.

### C.3.3 Contrastive Losses

In this section, we present an ablation on the effect of the contrastive losses that are applied in real world datasets. The experiments are performed on a slightly modified version of CIFAR-10 and shown in Table 6. As can be seen, the experiments showed that the proposed combination of (A) and (B) leads to the best results as the two methods complement each other.

### C.3.4 Unknown number of clusters

In an unsupervised setting, it is unclear a priori what the optimal number of clusters is. Thus, we are interested in the sensitivity of TreeVAE with respect to misspecifications of the ground truth number of clusters. The CIFAR-100 dataset is an optimal candidate for this ablation, as the ground truth number of clusters can be defined differently depending on whether we look at coarse or fine-grained classes. As a metric, we will only use NMI, because DP, LP, as well as Accuracy values are not fairly comparable across different number of predicted clusters. To determine the sensitivity of TreeVAE with respect to the misspecification of number of clusters, we evaluate its performance with 10, 20, and 32 predicted classes, compared to the ground truth 20 superclasses. The results are depicted in Table 8. Clearly, TreeVAE is not very sensitive to the correct specification of the number of clusters. This is also supported by Fig. 10/11, which show that even as we grow the tree further than the ground truth number of classes, TreeVAE still finds meaningful subsets of inside the classes.

Table 7: Ablation study of contrastive losses. (A) corresponds to the NT-Xent regularization on the routers, (B) to the NT-Xent regularization on the bottom-up embeddings, (C) on a transposed version of the NT-Xent regularization on the routers, following Li et al. (2021), (D) to the NT-Xent regularization of only the output of the encoder, and (E) to minimizing the Jensen-Shannon divergence between the router probabilities of the two augmented inputs.

| Dataset | Contrastive Method | NMI |
|---|---|---|
| CIFAR-10* | Ours (=A+B) | $44.1 \pm 0.7$ |
| | (A) | $21.7 \pm 4.0$ |
| | (B) | $6.0 \pm 6.0$ |
| | (C) | $23.0 \pm 1.2$ |
| | (D) | $0.2 \pm 0.0$ |
| | (E) | $1.0 \pm 1.3$ |

Table 8: Test set hierarchical clustering performances (%) of TreeVAE for different number of predicted clusters. Means and standard deviations are computed across 10 runs with different random model initializations. The star "*" indicates that contrastive learning was applied.

| Dataset | # Predicted clusters | NMI |
|---|---|---|
| CIFAR-100* | 10 | $15.99 \pm 0.85$ |
| | 20 | $17.80 \pm 0.42$ |
| | 32 | $19.33 \pm 0.41$ |

As a next ablation, we are interested in the case of the number of ground truth clusters being misspecified or unclear. Note that the difference to the previous ablation is whether the ground truth or the predicted number of classes is misspecified. Again, CIFAR-100 is a natural choice for this ablation, as the ground truth number of classes can either be 20 or 100. Thus, in Table 9, we present how the performance of TreeVAE would change if we chose a different granularity of ground truth classes while keeping the number of predicted clusters fixed to 20. We can observe that the NMI for finer granularity is even higher than for the superclasses used, indicating that TreeVAE learned to differentiate classes of the same superclass.

# D   Datasets

This section provides supplementary background information and preprocessing steps of the different datasets used in our experiments.

## D.1   MNIST

The MNIST (LeCun et al., 1998) dataset is a widely used benchmark dataset in machine learning. It is composed of grayscale images representing handwritten digits from zero to nine. Each image is of size $28 \times 28$ pixels that we rescale to $[0, 1]$. The dataset is balanced, meaning that there is an equal number of examples for each digit class, with a total of $60'000$ training images and $10'000$ test images. The digits in the dataset are manually labeled, providing ground truth information for clustering evaluation.

## D.2   Fashion-MNIST

The Fashion-MNIST (H. Xiao et al., 2017) dataset consists of a collection of grayscale images depicting various clothing items and accessories. The dataset contains a balanced distribution of ten different classes, representing different fashion categories such as T-shirts, trousers, dresses, footwear, and more. Each image in the dataset has a resolution of $28 \times 28$ pixels that we again rescale to $[0, 1]$. Similar to the original MNIST dataset, Fashion MNIST also includes $60,000$ training images and $10'000$ test images.

Table 9: Test set hierarchical clustering performances (%) of TreeVAE for different number of ground truth clusters. Means and standard deviations are computed across 10 runs with different random model initializations. The star "*" indicates that contrastive learning was applied.

| Dataset | # Ground truth clusters | NMI |
|---|---|---|
| CIFAR-100* | 20 | $17.80 \pm 0.42$ |
| | 100 | $22.74 \pm 0.36$ |

## D.3 20Newsgroups

The 20Newsgroups (Lang, 1995) dataset is a widely-used benchmark dataset in the field of natural language processing and text classification, known for its inherent hierarchical structure. It consists of approximately $20'000$ newsgroup documents organized into 20 different topics or categories. These categories include sports, politics, technology, science, religion, and more. One of the most important characteristics of this dataset is the presence of hierarchical relationships within the topics. We employ a TF-IDF vectorizer for the $2'000$ most frequent words and use this text embedding as input to our model. We retain $60\%$ of the original dataset for training and separate the other $40\%$ for the test set.

## D.4 Omniglot

The Omniglot Lake et al. (2015) dataset contains handwritten characters from 50 different alphabets. Every alphabet contains various individual characters, amounting to a total of 1623 different characters in the dataset. Given the heterogeneous nature of the dataset, Omniglot contains relatively few samples. The dataset consists of $32'460$ samples, which is about 650 samples per alphabet or 20 samples per character. In our experiments, we are interested in whether TreeVAE can find alphabet-specific patterns that stay invariant across different characters. Given that many alphabets developed either from or parallel to each other, many alphabets are very hard to distinguish, both for humans and machines. While we use the full Omniglot dataset for our generative experiments to have more data, we only use a subset of the alphabets for our clustering experiments. We call this subset Omniglot-5 and it consists of the five alphabets Braille, Glagolitic, Cyrillic, Odia, and Bengali. We selected them by ensuring that we have languages from different origins (artificial, Slavic, and Indian) and that there are learnable hierarchies. Old Church Slavonic (Cyrillic) developed from Glagolitic and Odia and Bengali are languages with their own alphabet from geographically similar regions in India. The resulting Omniglot-5 results in $4'160$ samples. For both dataset versions and all experiments, we split the dataset into train/test splits with $80\%/20\%$ of the samples, respectively, and stratifying across the characters of the dataset. We resized the images to $28 \times 28$ to align with images in MNIST and Fashion-MNIST, transformed the images to grayscale, and rescaled pixel values to $[0, 1]$. For a description of the augmentations we use, we refer to Appendix E.1.2.

## D.5 CIFAR-10

The CIFAR-10 (Krizhevsky & Hinton, 2009) dataset contains colored images of ten balanced classes. These classes are $\{Airplane, Automobile, Bird, Cat, Deer, Dog, Frog, Horse, Ship, Truck\}$. Therefore, some classes are more similar (for example Automobile and Truck) than others (for example Frog and Airplane), indicating that there is a structure across clusters that can be captured. The dataset consists of $50'000$ training and $10'000$ test images of resolution $32 \times 32$ pixels and 3 color channels, which we rescaled to $[0, 1]$.

## D.6 CIFAR-100

The CIFAR-100 (Krizhevsky & Hinton, 2009) dataset originally contains colored images of 100 classes. In order to not have to grow until TreeVAE has 100 clusters, we apply the common grouping of the 100 classes into 20 balanced superclasses. In Table 9, we also present an ablation if instead we chose the full 100 clusters. The dataset consist of $50'000$ training and $10'000$ test images of resolution $32 \times 32$ pixels and 3 color channels, which we rescaled to $[0, 1]$.

### D.7 CelebA

The CelebA (Z. Liu et al., 2015) dataset contains images of celebrities with 40 attribute annotations per image. This dataset is well-suited for exploratory analysis, as there are no agreed-upon ground truth clusters. As a result, we can visualize the clusters that are learned by TreeVAE and validate whether they align with our intuition as well as with the given attributes. Thus, in Table 3, we additionally define "Hair Loss" as balding or having a receding hairline, and "Beard" as having a beard or the so-called 5 o'clock shadow. For training, we select a subset of $100'000$ random images from the training set and evaluate on the given test set of $19'962$ images. Lastly, we crop the images to $64 \times 64$ pixels with 3 color channels and rescale them to $[0, 1]$.

## E  Implementation Details

The following section provides a comprehensive overview of the various aspects and components regarding the practical implementation of our proposed framework. We provide a detailed description of our proposed model's training process and architectural design. Furthermore, we outline the specific techniques and methodologies used for the contrastive learning extension. We also provide the implementation details of the Variational Autoencoder and the Ladder Variational Autoencoder that are used as baselines. Finally, we present the computational resources required for executing the proposed framework. Together, these subsections aim to provide a comprehensive view of the implementation details, enabling readers to replicate and build upon our work effectively.

### E.1  Training Details

Hierarchical variational models, such as LadderVAE, are prone to local minima and posterior collapse, and TreeVAE shows similar behavior. Therefore, few technical choices, such as batch normalization and KL annealing, are needed to converge to a desirable optimum. For all datasets, we train TreeVAE only on the training set and evaluate the trained model on the separate test set.

#### E.1.1  Training the Tree

For MNIST, Fashion-MNIST, Omniglot, and 20Newsgroup, the trees are trained for $M = 150$ epochs at each growth step, and the final tree is finetuned for $M_f = 200$ epochs. To reduce the risk of posterior collapse during training, we anneal the KL terms of the ELBO. Starting from 0, we increase the weight of the KL terms every epoch by $0.001$ except for the final finetuning of the full tree. Here, we linearly increase the weight by $0.01$ until we reach 1, such that the final model is trained on the complete ELBO. Additionally, for the aforementioned datasets, we finetune the full tree after every third growing step for 80 epochs with KL annealing of $0.001$, so the otherwise frozen parameters can adapt to the new structure. For CIFAR-10, CIFAR-100, and CelebA, the splits are mostly determined by the contrastive losses, and therefore, we avoid finetuning the full tree to reduce the computational time. The KL term is annealed every epoch by $0.01$ such that the weight reaches 1 after 100 of the 150 epochs every growth step. While the annealing procedure is required to achieve good performances, the exact annealing decay plays a smaller role in the final performances, as well as the in-between finetuning of the trees, and the number of epochs at each training step.

#### E.1.2  Augmentations

Due to the scarcity and heterogeneity of Omniglot, we trained our models using data augmentation by randomly rotating images by up to $\pm 10$ degrees, horizontally/vertically shifting images by up to $\pm 2$ pixels, applying up to $1\%$ shearing, and zooming by up to $\pm 10\%$. Note that for Omniglot, no contrastive learning was applied; the augmentations were solely used for overcoming the scarcity of data points.

On the other hand, in CIFAR-10, CIFAR-100, and CelebA, augmentations were applied to use contrastive learning as introduced in Section 2.6 and Appendix E.3. For CIFAR-10 and CIFAR-100, first, each image is randomly cropped and resized with scale $\in (0.2, 1)$ and aspect ratio $\in (3/4, 4/3)$. Second, each image is randomly flipped in the horizontal direction with probability 0.5. Third, with probability 0.8, color changes are performed, which consist of changes in brightness, contrast, and saturation, all with their respective factors $\in (0.6, 1.4)$, as well as change in hue with its parameter

$\in (-0.1, 0.1)$. Lastly, with probability $0.2$, we transform the image to grayscale. The chosen parameters are largely adopted from Li et al.'s (2022) weak augmentation scheme.

While these augmentations lead to superior group separation, in CelebA, we are also interested in generative results. Therefore, to obtain generations without non-sensical artifacts, we reduce the augmentation strength by removing the hue and grayscale augmentations, reducing the other color parameters by a factor 2, and reducing the cropping parameters to scale $\in (3/4, 1)$ and aspect ratio $\in (4/5, 5/4)$.

### E.2   Model Architecture

In general, model architectures for all datasets are similar and consist of an encoder, followed by the tree structure with the transformations, routers, and the bottom-up, as depicted in Figures 2 and 3, and identical decoders for each leaf of the tree. We always apply batch normalization followed by Leaky ReLU non-linearities after all convolutional and dense layers. For further details on the specific implementation of the sections below, we direct the reader to consult the accompanying code.

#### E.2.1   Transformations/Bottom-Up

The architecture of the transformations and the bottom-up is identical across all experiments. They take as input the previous latent vector of dimension $4$ (20Newsgroup), $8$ (MNIST, Fashion-MNIST, Omniglot), or $64$ (CIFAR-10, CIFAR-100, CelebA) and map them to some intermediate representation of dimension $128$ (20Newsgroup, MNIST, Fashion-MNIST, Omniglot), or $512$ (CIFAR-10, CIFAR-100, CelebA) using a dense layer, followed by a dense layer without activation to compute $\mu$ and one with Softplus activation to compute $\sigma$ of the approximate posterior/conditional prior of the respective tree level.

#### E.2.2   Routers

Similar to transformations, the architectures of routers are identical across all experiments. The generative routers take as input $z_i$, the latent representation of a node $i$, while the inference routers use $\mathbf{d}_{depth(i)}$. Then, both sequentially apply two dense layers that both map them to $128$ (20Newsgroup, MNIST, Fashion-MNIST, Omniglot), or $512$ (CIFAR-10, CIFAR-100, CelebA) dimensions, after which a final dense layer with a Sigmoid activation function computes the probability of the binary decision that infers the next child node.

#### E.2.3   MNIST/Fashion-MNIST

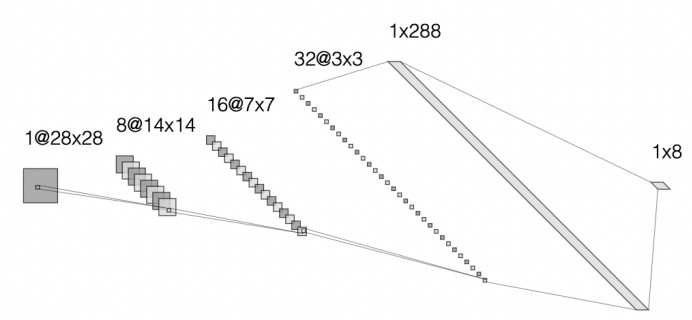

Figure 18: MNIST and Fashion-MNIST encoder architecture.

For MNIST and Fashion-MNIST we use identical encoder/decoder architectures. Both datasets have inputs of dimension $28 \times 28$ and 8 dimensional latent spaces. We use a small encoder as depicted in Figure 18, and a symmetric decoder. The encoder uses $3 \times 3$ convolutions with stride 2, whereas the decoder uses transposed convolutions.

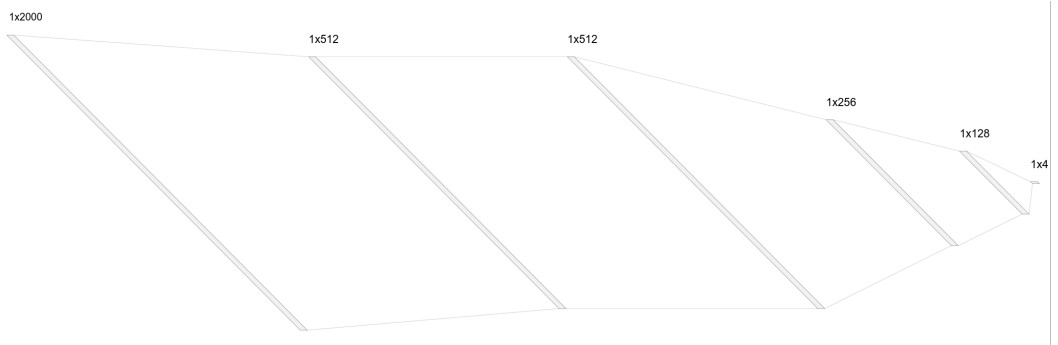

Figure 19: 20Newsgroups encoder architecture.

### E.2.4 20Newsgroups

20Newsgroups is a text dataset, so the encoder/decoder architecture differs from the other experiments. The samples of 20Newsgroups are preprocessed in such a way, that the input becomes a 2000D vector (see Section D.3). Hence, the encoder for 20Newsgroups is a simple MLP with 5 dense layers, as depicted in Figure 19, and the decoder again mirrors the encoder architecture.

### E.2.5 Omniglot

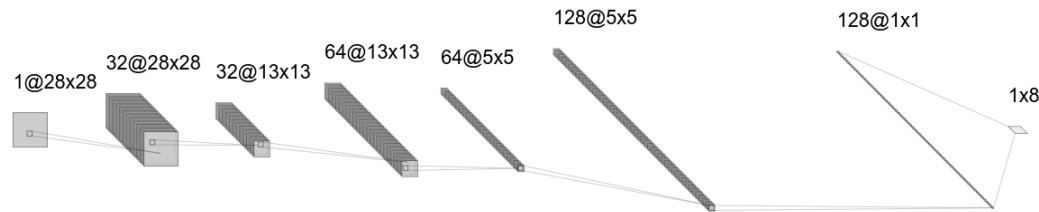

Figure 20: Omniglot encoder architecture.

For Omniglot, we use an architecture similar to the one for MNIST and Fashion-MNIST. In contrast to Section E.2.3, we increase the receptive field of the encoder by using $4 \times 4$ convolutions and make it 4 times wider and 2 times deeper. We refer to Figure 20 for a schematic of the encoder. As in the other architectures, the decoder is symmetric to the encoder.

### E.2.6 CIFAR-10/CIFAR-100/CelebA

For real-world image datasets, more complex architectures are required. We first define a ResnetBlock as two consecutive convolutions with kernel size 3 and stride 1, whose output gets weighted by 0.1 and combined with a residual connection to the input of the first convolution. In case input and output dimensions do not match, the residual connection consists of a convolution with kernel size 1 and stride 1 to match dimensionality. The encoder consists of consecutive ResnetBlocks with 2D average-pooling of kernel size 3 and stride 2. Here, number of filters per ResnetBlock are increasing by a factor of 2, starting from 32 and going up to 256 until the representation is $4 \times 4 \times 256$, at which point we flatten it to get $\mathbf{d}_L$. We refer to Figure 21 for a schematic of the encoder. The decoder structure is similar to the reversed encoder, where the pooling layers are replaced by 2D upsampling with bilinear interpolation and before the ResnetBlocks, we pass the input through a dense layer and consecutive reshaping in order to match the expected input dimensions. Slight differences are required for CelebA due to the higher resolution. To arrive at the same internal representation, we add one more ResnetBlock with 256 filters to the encoder, as well as one more ResnetBlock at the start of the decoder, starting from 16 instead of 32 filters.

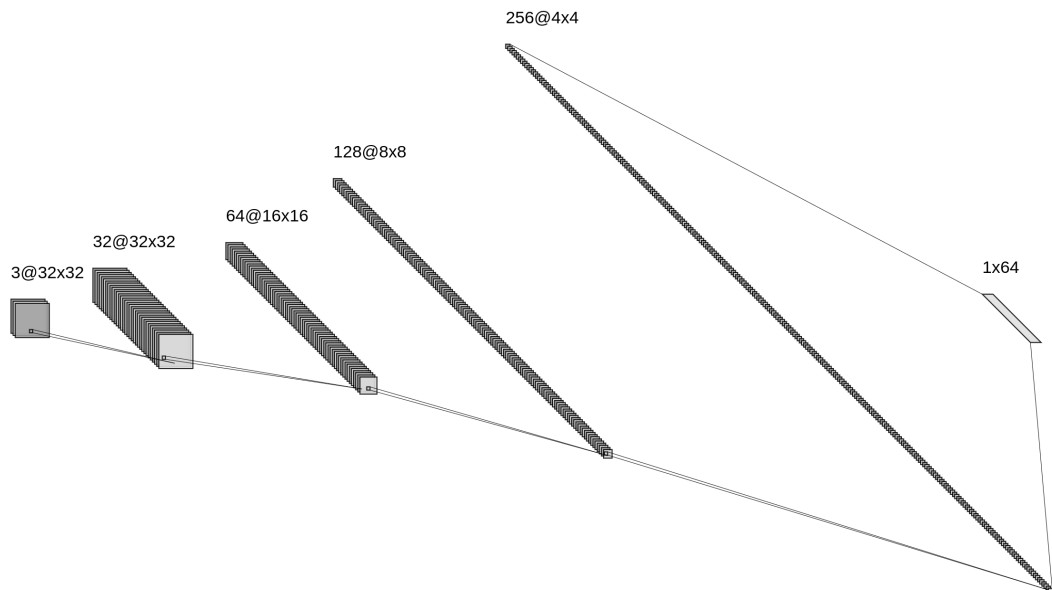

Figure 21: CIFAR-10, CIFAR-100, and CelebA encoder architecture. The initial convolutional layer comprises a ResnetBlock, while the remaining convolutional layers additionally incorporate average pooling prior to the ResnetBlock.

### E.3 Contrastive Learning

In real-world datasets, data splits, as determined by the ELBO and especially by the reconstruction loss, might not coincide with the human-aligned notion of clusters. For example, we observed that plain TreeVAE utilizes specialized decoders to focus on good reconstructions of the various background colors. These data splits are rather intuitive from a reconstruction point of view, however, these might not encode contextually meaningful splits. Thus, we introduce an additional contrastive loss to guide TreeVAE toward meaningful clusters. The idea of contrastive learning (Sohn, 2016; van den Oord et al., 2018; Wu et al., 2018; Chen et al., 2020) is simple; constrain the space of potential clustering rules by defining certain data augmentations with respect to which the model's cluster assignment should become invariant. The model should assign a sample to the same cluster independent of whether and which subset of the chosen augmentations are applied to it. For the augmentations that were used for contrastive learning, we refer to Appendix E.1.2.

To achieve the desired behavior, we utilize the normalized temperature-scaled cross-entropy loss (NT-Xent) (Sohn, 2016; van den Oord et al., 2018; Wu et al., 2018; Chen et al., 2020), whose definition we repeat here for ease of reading: $\ell_{i,j} = -\log \frac{\exp(s_{i,j}/\tau)}{\sum_{k=1}^{2N} \mathbb{1}_{[k \neq i]} \exp(s_{i,k}/\tau)}$, where $s_{i,j}$ denotes a similarity measure, which we define as the cosine similarity between the representations of $\tilde{x}_i$ and $\tilde{x}_j$, and $\tau$ is a temperature parameter. This is computed for all $2N$ pairs of augmented samples $\tilde{x}_i$ and $\tilde{x}_j$ originating from the same initial sample $x$ and then averaged. We compute this depth-wise for every embedding $\mathbf{d}_l$ by first passing it through a separate MLP with one hidden layer of dimension 512, followed by batch normalization and LeakyReLU and the final linear layer with output dimension 64, on which the NT-Xent with $\tau = 0.5$ is applied. Then, we average the pairwise losses and sum them up over all $\mathbf{d}_l$, where we, to be invariant to the depth of the bottom-up, divide by the maximum depth. With this, we regularize the bottom-up embeddings learned by the model to encode every augmented pair of samples close to each other, thus constraining the learning of information that the model should be invariant to.

Even with the previous regularization, $\mathbf{d}_l$ might still contain some of information that is not relevant for clustering (such as background color), and therefore the routers might base their split on such characteristics. Therefore, we build upon ideas from Li et al. (2021, 2022), which propose not only to regularize the embeddings, but also the cluster probabilities themselves. While Li et al. (2021, 2022) change from a sample-wise view to a cluster-wise view for calculating the NT-Xent on the $2K$ clusters, we instead keep the sample-wise view and apply the NT-Xent with $\tau = 1$ directly on the

outputs of every bottom-up router $q(c_i \mid \mathbf{d}_l)$. We would like to emphasize here that we do not project them to another space, because, in contrast to previous methods (Chen et al., 2020), we wish to remove all irrelevant information with respect to the augmentations from the routers. The computed pairwise loss is then weighted by the probability of falling into the given node. Lastly, we sum up the weighted pairwise losses and divide them by the sum of the weights to ensure equal regularization in every node. This has three advantages: (*i*) We can use the proposed loss during the training of the subtrees, thus gaining computational efficiency. (*ii*) We implicitly encourage the splitting of the data into balanced classes, following the overall design choice for splitting the data, outlined in Section 2.5. (*iii*) We can omit the computationally intensive finetuning of the full tree, as the splits, as enforced by the contrastive losses, are local, which alleviates the need of finetuning the full tree for global optimization.

Lastly, we add these two contrastive parts together and weigh them by $100$ to match the gradients of the ELBO. Thus, we naturally introduce a tradeoff between reconstruction quality and clustering quality, as we constrain the model to a clustering solution that does not optimally make use of specialized decoders.

### E.4 Baselines

For a fair comparison, we retain the same architectural choices made for TreeVAE for both the Variational Autoencoder and Ladder Variational Autoencoder, used as baselines. In particular, we retain the same encoder and decoder structure as explained in Section E.2. For datasets on which TreeVAE applies contrastive learning, we similarly apply the regularization on the bottom-up embeddings of the baselines. Below we describe specific training details that deviate from TreeVAE, due to the differences in the graphical models.

#### E.4.1 Variational Autoencoder

The non-hierarchical counterpart of TreeVAE is a classical Variational Autoencoder (VAE). As it does not contain a hierarchical structure, the VAE consists solely of an encoder and decoder architecture, which follows the architectural choices of TreeVAE. We train the VAE for $500$ epochs and use the KL annealing procedure proposed by LadderVAE, which linearly increases the weight of the KL terms from $0$ to $1$ in $200$ epochs.

#### E.4.2 Ladder Variational Autoencoder

The sequential counterpart of TreeVAE is the LadderVAE (Sønderby et al., 2016). We retain the same encoder, decoder, and transformations used by TreeVAE, and we set the depth to $5$ for all datasets. We train the LadderVAE for $1000$ epochs and use the KL annealing procedure proposed by the authors, which linearly increases the weight of the KL terms from $0$ to $1$ in $200$ epochs.

#### E.4.3 Agglomerative Clustering

We train Ward's minimum variance agglomerative clustering (Ward, 1963; Murtagh & Legendre, 2014) on the plain input space (Agg), the latent space of the VAE (VAE + Agg) and the last layer of LadderVAE (LadderVAE + Agg). We observed that clustering on the last layer of LadderVAE performed substantially better than when using the root (or the top layer). We chose Ward's method as it performed better than other classical hierarchical clustering methods (such as single-linkage agglomeration clustering or bisecting K-means). However, Ward's method cannot be tested on a held-out dataset, as it inherently does not allow for prediction on new data. For this reason, we train and test Ward's method on the test set embeddings given by the baseline models, which were trained on the training data, for a fairer comparison with the other methods.

### E.5 Resource Usage

In the following, we describe the resource usage of our experiments. All our experiments were run on RTX3080 GPUs, except for CelebA, where we increased the memory requirement and use a RTX3090. Training TreeVAE with 10 leaves on MNIST, Fashion-MNIST, and Omniglot-50 takes between 1h and 2h, Omniglot-5 30 minutes, CIFAR-10 5h. Training TreeVAE with 20 leaves on 20Newsgroup takes approximately 30 minutes, and on CIFAR-100 9h. Training TreeVAE on CelebA

takes approx 8h. Please note that we only report the numbers to generate the final results but not the development time.

