# OpenReview forum: "Tree Variational Autoencoders"
_NeurIPS.cc/2023/Conference — NeurIPS 2023 spotlight_

### Official Review · Reviewer_822K · 2023-07-03

**Soundness:** 3 good
**Presentation:** 4 excellent
**Contribution:** 4 excellent
**Rating:** 8
**Confidence:** 4

**Summary:**

This paper introduces a new class of variational autoencoders that define a binary tree structure. Given a fixed tree, the generative model is a top-down hierarchical model with binary routing components and the inference network follows a top-down approach in the style of the Ladder VAE. The authors propose to grow the tree iteratively every N training steps. They also suggest using `NT-Xent` to force aligning the learned representation with human perception.

Based on binary image, natural image and a text dataset, the authors demonstrate the effectiveness of the method based on modelling (likelihood) and clustering performances. Examples of learned trees are presented in the paper and are convincing: coupled with `NT-Xent`, Tree VAE learns convincing data hierarchies without explicit supervision.


**Strengths:**

A strong, creative and well-written paper!

1. This paper makes a significant contribution towards learning meaningful hierarchical models, which is an important problem in the field
2. The solution described in this paper is creative and well presented (structured presentation: generative model vs. inference network, detailed derivations, intuitive notation)
3. The experiments are extensive (5 datasets, 4 baselines, 10 seeds for each run)
4. The visual results are very convincing
5. In-depth appendix

**Weaknesses:**

1. Section 2.6 is unclear as such. I suggest adding the equation corresponding to the auxiliary loss term that comes with the use of `NT-Xent`
2. The high quality of the results might be tightly tied to the use of `NT-Xent` .  According to appendix E.3, a lot of work is required to get this right. I think this work would greatly benefit from an ablation study targeted at the use of choice of methods for integrating prior knowledge (e.g., without `NT-Xent`, with another method)

**Questions:**

1. What is the modelling performance (likelihood) of Tree VAE without `NT-Xent`?

**Limitations:**

Limitations of the method are not sufficiently discussed. The authors might want to discuss some of the following points more thoroughly
1. the multi-stage training of TreeVAE adds complexity, and result might vary depending on the choice of expansion criterion
2. the results depend of the choice of "prior knowledge" model

---

> ### Author Rebuttal · Authors · 2023-08-09
>
> Thank you for your comments and feedback! We appreciate your support for this paper.
>
> W: We will make sure to extend Section 2.6 for the camera-ready version. We intentionally kept it short, as we only use `NT-Xent` for real-world image datasets, that is CIFAR-10, CIFAR-100, and CelebA. Here, the reconstruction loss is not sufficient for guiding the tree toward a meaningful clustering, which is why for those datasets, we inject inductive biases through contrastive learning. As such, the results of MNIST, FMNIST, 20Newsgroup, and Omniglot are solely tied to the maximization of the ELBO and the presented generative modeling performance of TreeVAE is without the `NT-Xent`. For the choice of method for integrating prior knowledge, we have experimented with multiple options. They are all inspired by the fact that two augmented versions of the same input should behave similarly. Please note that the following results were obtained on a curated subset of CIFAR-10 as this ablation was performed during the model development stage. Thus, the absolute values are not directly comparable to the values in our paper, however, the relative differences in methods remained comparable over different experiments, so we believe, in order to save computational resources, that these results are still meaningful.
> |Method|NMI|
> |-|-|
> |Ours (=A+B)|$44.1 \pm 0.7$|
> |(A)|$21.7 \pm 4.0$|
> |(B)|$6.0 \pm 6.0$|
> |(C)|$23.0 \pm 1.2$|
> |(D)|$0.2 \pm 0.0$|
> |(E)|$1.0 \pm 1.3$|
> Here, (A) corresponds to the `NT-Xent` regularization on the routers, (B) to the `NT-Xent` regularization on the bottom-up embeddings, (C) on a transposed version of the `NT-Xent` regularization on the routers, following Li et al. [2], (D) to the `NT-Xent` regularization of only the output of the encoder, and (E) to minimizing the Jensen-Shannon divergence between the router probabilities of the two augmented inputs. Further experiments showed that the combination of (A) and (B) leads to the best results.
>
> Q: We hope that we were able to answer your question in the previous section noting that the modelling performance in the submission is already without `NT-Xent`.
>
> L1: Lastly, regarding the multi-stage training, we agree with the reviewer and leave the investigation of a more principled approach regarding the expansion criterion to future work. We will make sure to include this in the limitations section of the camera-ready version. Additionally, we would also like to refer to our answer to Reviewer UPHe, where we show that given enough computational resources, the computational complexity is the same as for an iteratively grown LadderVAE.
>
> L2: Discussed in the weaknesses section.
>
> If you have any additional questions, comments, or feedback, we will be happy to address them.

---

### Official Review · Reviewer_NuUb · 2023-07-05

**Soundness:** 3 good
**Presentation:** 3 good
**Contribution:** 3 good
**Rating:** 6
**Confidence:** 3

**Summary:**

The paper presents a novel method of unsupervised hierarchical clustering by encoding structural sequential dependencies between hidden variables within the framework of variational auto-encoders. The authors adopt similar designs of top-down and bottom-up dependency structure to Ladder VAE, but imposes a binary tree or decision tree based structure to prior and posterior distribution of latent variables instead of fixed chain structure adopted by Ladder VAE, to discover hierarchical and well-organized dependencies among encoded representations. In the proposed decision tree structure, all hidden variables are organized in a top-down architecture with a nested set of binary gating/decision variables controlling the direction of every move, and altogether optimized with a unified objective function derived through variational inference. To make the binary tree structure flexible and learnable, a sequential tree growth procedure is also developed. The paper compares TreeVAE with both non-hierarchical and hierarchical baselines, and it shows superior clustering performance and achieves competitive log-likelihood lower bound.

**Strengths:**

1) The paper is well written, and is easy to follow. Informative figures and detailed formulae are given to explain how the model is built, which is very helpful.
2) Although the perspective of modeling hierarchical clustering as a binary decision problem is not novel, the idea of extending Ladder VAE to the hierarchical clustering task is interesting and  looks natural.
3) Quite a few illustrative examples are introduced to validate the meaningful hierarchical structures learned by TreeVAE (e.g., Figure 1 and Figure 4), which makes the paper easier to follow and the results more convincing.
4) Extensive experiments are conducted on multiple datasets (e.g., MNIST, Fashion, 20Newsgroups, Omniglot, CIFAR-10/100, CelebA) and multiple tasks (e.g., hierarchical clustering, sample generation and hierarchy discovery) and with quantitative results compared with state-of-the-art baseline methods.


**Weaknesses:**

1) The paper provide comprehensive experimental results, which is good. But The qualitative results are not explained very clearly, prompting people to doubt if the proposed method can really generate proper hierarchies that is consistent with the dataset’s intrinsic structure.
2) Some important ablation studies are missing, such as data augmentation and the pre-determined maximum depth of the binary tree.
3) For a generative model, sample generation should be an important part of the evaluation, but the implementation and experimental results are unclear in the context and mostly postponed to the appendix, making it quite confusing.
4) Since the tree growth procedure is proceeded in an iterative and sequential way, the corresponding training efficiency is doubtful. And even so, before the procedure we still have to set up the maximum depth of the tree or the number of leaves depending on specific dataset.
5) Data augmentation is a relatively weak form of prior knowledge especially for the hierarchical clustering task, it would be appreciated to incorporate other easily accessible forms if possible.


**Questions:**

1) The pre-defined maximum depth of the binary tree or number of total leaves should be a major factor for hierarchical methods. So it would be more convincing to take different choices of this factor into account and make comparisons accordingly.
2) Could the authors explain how the labeled hierarchical structure shown in Figure 5 is obtained? If it is an empirical summary, the results will be not convincing enough since there might exist mismatch between ground-truth labels and the clustering hierarchies learned by the model, with clustering accuracies less than 50 percent as shown in Table 1.
3) The results of Figure 6 and Table 3 are really confusing and vaguely explained. What does the term ‘leaf-frequency’ mean and what can be concluded from the results? Could the authors explain it in a clearer way?
4) Since the binary tree structure is learned in a sequential way, would it be better to term it a ‘suboptimal’ or ‘locally optimal’ tree instead of ‘optimal’ tree adopted in the paper?


**Limitations:**

see above

---

> ### Author Rebuttal · Authors · 2023-08-09
>
> Dear reviewer NuUb,
> thank you for your feedback and constructive criticism.
>
> W1: Due to the space limitations, we have refrained from explaining the qualitative results in greater detail, however, we will make sure to include this in the camera-ready version. There are also additional figures in Appendix C, which we described in more detail. We also want to point toward Fig. 1 of the additional PDF combined with our response to reviewer bo43, where we perform an analysis of the root embedding of FMNIST. We show that the in the learned embeddings, bags are embedded in between shoes and tops, while trousers are an entirely separate category for themselves. Additional evidence is presented in the 20Newsgroup dataset, which we talk about in your Q2 below.
>
> W2: Additionally to the ablation studies in Appendix C, We have added an extensive ablation study in Table 1 of the additional PDF, as well as presented a comparison of different contrastive approaches in our response to reviewer 822K. We want to emphasize that without data augmentation, the clustering performance for the CIFAR datasets would lower greatly, as the model clusters according to background information, which usually influences reconstruction loss more than the foreground information.
>
> W3: Thank you for this comment. As we are solving two tasks in one (those being generation and hierarchical clustering), we decided to emphasize the discovery of hierarchies in the main text. We will make sure to move selected generative figures from the Appendix into the main text for the camera-ready version.
>
> W4: We have experimented with multiple growing procedures and the iterative way has proven to be the most stable one, see Fig. 2 of the additional PDF for a visualization of the training ELBO behavior on FMNIST. By iteratively growing, we divide the complex problem into simpler subproblems, which is inspired by agglomerative clustering. Additionally, with enough computational resources, one could grow all nodes of the same depth simultaneously, as the resulting subtrees are conditionally independent, which would further speed up training. Note also that during training of the subtree, the weights of the rest of the tree are frozen (see Fig. 3 (right) of the main paper), which improves training efficiency.
> Regarding the stopping criterion of the clustering procedure, we want to mention that it is common to set the number of clusters $K$ a priori. Nevertheless, we agree that this is unsatisfactory and will focus future efforts on finding smarter ways in going about this issue, as well as mention it in the Limitations section of the final version.
>
> W5: A goal of our method is to perform hierarchical clustering with as little inductive biases as possible, in order to be truly unsupervised and also stay true to maximizing the evidence lower bound. Therefore, we only add additional supervision in the weak form of data augmentation to guide the model towards desirable splits, when it is required, which is the case for the CIFAR datasets and CelebA. Naturally, future extensions can be made where stronger inductive biases can be leveraged in order to guide and support the clustering procedure, however, this is not the focus of our current work, as we want to put focus on the fact that the model itself already works very well. This is, we propose a theoretically founded model class, where the contrastive extension shows that combining it with forms of supervision is possible.
>
> Q1: We refer to Table 1 of the additional PDF, where we have conducted an ablation analysis for different pre-defined maximum depths of the tree while keeping the maximum number of leaves $K$ constant. Additionally, the metrics Dendrogram Purity (DP) and Leaf Purity (LP) in the main paper allow unbiased evaluations for dendrograms where the number of learned clusters different from the ground-truth number, which is why we compute these two metrics with trees that contain $20$ leaves. Lastly, we also want to mention the further experiments in Appendix C, where we depict trees that are grown to more than $10$ clusters, as well as quantitative results for varying the number of ground-truth clusters, as well as the number of learned clusters.
>
> Q2: 20Newsgroup consists of $20$ classes that also have a hierarchical structure. The labels depicted in the figure are the $20$ ground-truth labels of the dataset, where we visualize the majority class of the samples falling into each leaf. The words in each class label indicate the group assignments on each hierarchical level, where each level is separated by a dot. For example, "comp.graphics" indicates a first hierarchical layer of "computer" and a second layer of "graphics". To be clear, we did not choose the hierarchical level of each class in Fig. 5; it is predetermined by the creators of the dataset. Our experiments show that TreeVAE discovers the correct hierarchies, which supports the claim that we uncover meaningful hierarchies.
>
> Q3: The CelebA experiments are performed to give an alternative experiment to simply recovering already known labels. Here, we let the model grow and try to interpret the results in a more explorative fashion. We will make sure to put more information in the camera-ready version. Every number $v_{a,k}$ in Table 3 indicates for a given attribute $a$ and a given leaf $k$ the percentage of all samples that fall in $k$, in which attribute $a$ is present. Thus, we observe that people with blonde hair mostly fall into cluster $8$ (given the knowledge that the number of samples is evenly distributed).
>
> Q4: Thank you for the suggestion, we agree with the pointed-out issue and will adopt "locally optimal" in the camera-ready version.
>
> We hope our general response and the response to your review address the established weaknesses and questions. We are happy to address any additional feedback or questions that might arise.

---

### Official Review · Reviewer_bo43 · 2023-07-05

**Soundness:** 3 good
**Presentation:** 3 good
**Contribution:** 3 good
**Rating:** 7
**Confidence:** 3

**Summary:**

introduces a new generative hierarchical clustering model called Tree Variational Autoencoders (TreeVAE) that uncovers hidden structure in data by adapting its architecture to discover the optimal tree for encoding dependencies between latent variables. The authors compare TreeVAE to other generative models and demonstrate its effectiveness in uncovering underlying clusters and hierarchical relations in real-world datasets. They also discuss the advantages and disadvantages of using TreeVAE, such as its ability to provide a tighter lower bound of the log-likelihood at the expense of a larger neural network architecture and an increase in the number of parameters. Overall, this paper presents an innovative approach to generative modeling that has the potential to improve our understanding of complex data structures.

**Strengths:**

1. The quality of exposition is commendable. The paper presents a clear narrative, beginning with the intuitive premise, followed by a comprehensive description of the model, including a highly useful algorithmic representation. The logical sequencing of these sections facilitates reader comprehension. The authors have succeeded in presenting a complex topic in an accessible and cogent manner.

2. The innovation presented through the proposition of a tree structure in the latent space of a Variational Autoencoder (VAE) constitutes a significant contribution to the field. This unique concept not only manifests novelty but also offers functional value as it endows the model with enhanced performance capabilities. The authors have adeptly demonstrated how this structural adjustment can add a new dimension to the potential applications of VAEs and its progenies.

3. robust set of qualitative and quantitative evaluations are delivered in the paper. The authors have not shied away from providing extensive empirical evidence to assert the superiority of their model. The data presented is both extensive and convincing, demonstrating the improved performance of their model compared to common benchmark models. The evaluative components of the paper are meticulously presented, reflecting the comprehensive and rigorous experimental protocols the authors have employed. The authors' due diligence in this aspect lends credibility to their claims and effectively underlines the practical implications of their research.



**Weaknesses:**

1. The authors have stated that the clusters learned by their model possess semantic meaning. However, it is prudent to query if this semantic characterization emerges from explicit feedback or training, or if it is simply a by-product of the model's intrinsic clustering tendencies. For instance, in Figure (4), items such as pants are grouped alongside shoes and bags, while being distinct from tops. This calls into question the semantic integrity of the learned representation. Hence, it would be advantageous for the authors to conduct an additional layer of embedding analysis, in order to ascertain the proximity of these items beyond mere qualitative observation. Such an endeavor would bolster the readers' understanding and lend more weight to the claim that the cluster learned is indeed semantic.

2. he paper could benefit from a more extensive ablation study that investigates the effect of different components and hyperparameters on the model's performance. For instance, an intriguing area of exploration could be the impact of modifying the number of layers in the tree. It would be worthwhile to examine if augmenting this parameter might encourage a more detailed clustering. A comprehensive study of this nature could elucidate the relative contributions of each component, making clear which are indispensable to the model's performance, and which are more auxiliary in nature. By adding more depth to the discussion of how these elements influence the final result, the authors would be equipping their audience with a more nuanced understanding of their model's operation.

These enhancements would provide further depth and rigor to the paper, effectively strengthening the authors' claims and helping readers to more fully grasp the novelty and impact of the presented research.


**Questions:**

Please refer to the above sections for detailed discussion.

**Limitations:**

Yes, there is a section at the end of the paper discussing limitations and future research opportunities.

---

> ### Author Rebuttal · Authors · 2023-08-09
>
> Dear reviewer bo43,
>
> Thank you for your comments and your thorough reading of the paper.
>
> W1: The idea behind the claim of semantic meaningfulness is that samples of the same cluster should have a similar latent representation and also that clusters that are close to each other in the hierarchical structure should have more similar latent representations than clusters far apart. This is integrated in the model design, as the learning of the decision boundary is mainly guided through the weighted reconstruction loss. That is, the split in every node is optimized such that the two decoders can specialize themselves as well as possible on the subgroup of data, and therefore the subspace of representations, they observe. Thus, the routers try to find splits that partition the data into two subgroups that are as distinct as possible while being as similar as possible within, which coincides with partitioning the latent space at every node into two semantically different subspaces. With this objective, our model should be able to recover semantically meaningful clusters through the training procedure. A direct way of checking this is by exploring the learned tree of the 20Newsgroup dataset in Fig. 5 of the main work. For each leaf, we denoted the majority class that falls into it. Note that the name of each class entails multiple hierarchical layers. As can be seen, the learned tree recovers the subgroups of different hierarchical levels very well, for example, all computer-related topics are grouped in one subtree. Thus, the human-assigned semantic structure corresponds to the unsupervisedly learned semantic structure. Furthermore, we have conducted an additional layer of analysis for the embeddings of FMNIST. Firstly, we have applied UMAP to reduce the root embedding to 2 dimensions and visualize them in Fig. 1 of the additional PDF. As can be seen, tops as well as shoes are clustered together, while bags are somewhat in between. Trousers on the other hand are completely separate. Our interpretation is that the root split is made between shoes and tops, where it is unclear in which subtree bags and trousers should fall as they are both not clearly assigned to one of the two groups. Therefore, depending on initialization, they might end up on either side. To analyze the effect of random weight initialization on the learned dendrogram, we have made further efforts to find a dendrogram that best summarizes the learned dendrogram of multiple runs. For this, we first align the leaves of the different trees by maximizing the overlapping samples, then we store the number of edges between any two clusters for every tree and then average this number over all trees. Thus, we have computed a distance matrix of all clusters, which is averaged over all trees. We then used average and complete linkage to cluster according to the average distance matrix. The recovered dendrograms of the two algorithms are identical, apart from the Bag and the Trouser cluster, which switch places. That is, all tops are in one subtree and all shoes in the other, while bags and trousers are assigned to either the shoes or tops subtree, depending on the clustering algorithm used. What this indicates is that the two clear groups of shoes and tops are consistently recovered, no matter the random initialization, while the assignment of bags and trousers varies, which is to be expected, as there is no clear right choice.
>
> W2: We agree that analyzing the effects of different components is important and have thus conducted an extensive ablation study presented in Table 1 of the additional PDF. Additionally, we would like to mention the further experiments performed in App. C, where for example we let the tree grow deeper, which lead to the discovery of different subgroups within the same class, for example, the differentiation of straight and tilted $9$'s or dark and light T-shirts.
>
> We hope our general response and the response to your review address the established weaknesses. We are happy to address any additional feedback or questions that might arise.

---

> > ### Comment · Reviewer_bo43 · 2023-08-21
> >
> > Thanks for the rebuttal. It answered all of the critical parts of my concerns. My rating would stay at "accept".

---

### Official Review · Reviewer_hJ8G · 2023-07-07

**Soundness:** 4 excellent
**Presentation:** 4 excellent
**Contribution:** 3 good
**Rating:** 7
**Confidence:** 5

**Summary:**

This paper introduces a new deep generative model, the tree variational autoencoder, designed to discover latent hierarchical clusters in data. The generative model makes a number of latent binary choices over a pre-learned tree structure, sampling a continuous representation at each node, then finally decoding the observed data point at the particular leaf of the binary tree that is reached. A scheme for performing variational inference on the model and a heuristic for learning the tree structure are given, and results presented for hierarchical clustering on common small image and text datasets.


**Strengths:**

Simple but novel technical idea for model and training, breaking it down into a number of simpler learning problems. Interesting qualitative results in terms of what hierarchy of clusters are discovered and strong qualitative results compared to baselines. Related work includes all relevant that I am aware of.


**Weaknesses:**

The datasets used seem quite small/simple in this day and age. I think it would be interesting to perform hierarchical clustering on the embeddings from a foundation model like Stable Diffusion on a much larger dataset. I think this may be what is being suggested in Line 318 in “Limitations & Future Work”. I would upgrade my score to Strong Accept if these results were present and favorable.

**Questions:**

Perhaps I’m missing something simple here but how is learning performed over the Bernoulli routers after the tree structure has been learned and all weights unfrozen? During the tree formation each learning problem has a single Bernoulli variable that is summed out but if the weights are unfrozen then aren’t there a combinatorial number of values to the Bernoulli variables?

**Limitations:**

Acknowledgments of a few limitations on Pg 9. I suspect there are specific implementation details where an alternative choice would make learning fail. It would strengthen the paper to have an ablation study where the sensitivity to model and training design choices are tested (i.e. going beyond the experiments in C.3).

---

> ### Author Rebuttal · Authors · 2023-08-09
>
> Thank you for your comments and your thorough reading of the paper.
>
> W1: We agree with this statement, as the purpose of this work is to introduce a new model class that jointly learns generation as well as hierarchical clustering. We are currently working on improving the model architecture, similar to how NVAE [3] improves upon LadderVAE, in order to reach SOTA performance on SOTA benchmarks, however, these results will not be available by the end of the rebuttal period and might justify a separate paper by themselves.
>
> Q1: During the training of the full tree, we sum over all $K$ paths to the leaves. Each term in the sum only depends on the realization within the path, that is, we can ignore all Bernoulli variables that do not lie in the path. Furthermore, for paths that share certain edges and nodes (think of two adjacent leaves), we only have to compute the intermediate values once, as they are shared between the paths. This reduces the computational complexity to being linear in the number of nodes.
>
> L1: We agree and refer to Table 1 of the additional PDF.
>
> If you have any additional questions, comments, or feedback, we will be happy to address them.

---

### Official Review · Reviewer_UPHe · 2023-07-08

**Soundness:** 3 good
**Presentation:** 3 good
**Contribution:** 3 good
**Rating:** 7
**Confidence:** 4

**Summary:**

This paper introduces a new architecture for variational auto encoders with a binary tree structured generative model. The approach takes the architecture of the Ladder VAE, but introduces a binary routing variable at each stochastic layer that allows generation to continue down one of two possible paths. The authors derive an evidence lower bound objective for their approach and a simple heuristic for learning the structure of the tree at training time. In their experiments the authors compare the clustering and generative modeling performance on a variety of benchmark image datasets.

**Strengths:**

This seems to be promising work and the model introduced in this paper does appear to have a number of advantages. The model is clearly introduced and the objective function derived by the authors seems sound to me, though I haven't gone through the details in depth. Unsupervised clustering of complex, high-dimensional data is a challenging and relevant problem. Both the quantitative and qualitative evaluations of the clustering performance seem to show that the method does find interesting structures in the data that correspond to known class labels, without having access to these labels at training time. The generative performance is also promising, though the gains are unsurprising given that the model can have substantially more parameters than its competitors. The method is also straightforward to implement and apply in practice.

**Weaknesses:**

While I think this work is interesting, I do have a number of concerns:
 - The computational complexity of training seems to be an issue, as computing the training objective requires evaluating every path in the tree. This likely means that this model is limited to very small trees and is not applicable to discovering large numbers of clusters. Similarly, the number of parameters is large for the depth. If clustering was not a concern it seems likely that the computational complexity would likely be better spent simply creating a larger hierarchical VAE model, though this matched parameter count comparison is not explored in the work. I do appreciate that for generation, the complexity only depends on the depth.
- I feel the work would benefit from further discussion and analysis of the tree-building routine. The proposed heuristic seems to work reasonably well, but is not well justified apart from the reasoning that the authors want to keep the tree balanced. This reasoning may fall apart if the true clusters are unbalanced. I would like to see discussion of how this choice interacts with the likelihood bound and whether the fact that some leaves can have significantly more generative capacity (as they are deeper in the hierarchy), has issues.
- The VAE/network architectures used for the experiments are somewhat out-of-date. The ladder VAE is substantially outperformed in generative performance by hierarchical VAEs like the NVAE or "very-deep" VAE of Child. State-of-the-art generative performance isn't necessarily the goal of this work, but on the chosen datasets the model presented is not competitive. The VAE + agg clustering baseline could similarly be updated to these newer (pertained) models or applied to something like a vector quantized VAE.
- The experiments are all on benchmark image datasets, which is fine, but I would be more interested in an application to a real-world dataset where the approach could provide new insights, rather than just rediscovering already known image labels.




**Questions:**

- How does the model perform if you start with a pre-specified tree structure?
- Equations 3 and 5 suggest that $q(z_i)$ is only dependent on the parent of $z_i$ and not $x$, which is inconsistent with equations 6-9 and the diagrams.
- Line 134, I'm unclear exactly what is meant by the routers having the same architecture as the generative model. Don't they depend on $x$?
- Equations 19 and 20 don't use $l$ anywhere within the summation, so it's unclear exactly how multiple samples are used.



**Limitations:**

The authors do discuss the limitations appropriately.

---

> ### Author Rebuttal · Authors · 2023-08-10
>
> We thank the reviewer for the thoughtful comments and feedback!
>
> W1: We agree that a naive training algorithm would have a heightened computational complexity, however, due to the structure of the model we can alleviate this issue: Firstly, we can store the values of every visited node such that we need to compute each only once, making it linear in the number of nodes. Secondly, the subtrees of all nodes with the same depth are conditionally independent, which implies that their computation can be parallelized. This reduces computational complexity to linear in the depth of the tree. Furthermore, during the greedy growing procedure, most of the tree is frozen, shown in Fig. 3 (right) of the submission. Lastly, even the greedy growing procedure can be parallelized, as subtrees are again conditionally independent. Thus, with enough resources available, we are able to recover the computational complexity of an iteratively grown version of LadderVAE, while simultaneously being able to cluster the data. We do appreciate that if not enough resources are available, one has to omit a few of the aforementioned speed-ups, however, we argue that this is the case for any parallelizable system. Nonetheless, we agree that the effective number of parameters \textit{during training} is higher for TreeVAE than for LadderVAE, with the notable difference that TreeVAE offers lightweight inference. Therefore, we have additionally conducted an experiment with matched the parameter count of LadderVAE. The results for LadderVAE are
> |Method|Acc.|NMI |LL |RL|ELBO|
> |-|-|-|-|-|-|
> |LadderVAE|$56.4 \pm 1.9$|$62.3\pm2.0$|$-233.7 \pm 0.1$|$224.8\pm0.3$|$-238.9\pm 0.3$|
> |TreeVAE|$63.6 \pm 3.3$|$64.7 \pm 1.4$|$-234.7 \pm 0.1$|$226.5 \pm 0.3$|$-239.2 \pm 0.4$|. As expected, this big LadderVAE is performing slightly better generatively, however at inference time, LadderVAE does not offer lightweight inference that reduces the effective parameter count to what is denoted in the main paper. We are quite satisfied that the differences, especially in the ELBO, are only marginal, while the significant difference in clustering performance stays similar. Note that FMNIST was chosen as the worst-case dataset for clustering performance, so we would expect the differences to be bigger in the other datasets.
>
> W2: Regarding the proposed tree-building routine, we took inspiration from decision trees for the iterative, greedy growing procedure. Our empirical experiments suggest that the number of samples as simple splitting criterion supports a consistent improvement of the ELBO, as can be seen in Fig. 2 of the additional PDF, where we monitored the ELBO after the convergence of every subtree.
> An alternative splitting criterion would be reconstruction loss, for which we conducted experiments that result in a slightly worse test ELBO of $-239.3 \pm 0.2$. Thus, we believe our criterion is a good starting point on which future work, e.g. combining our work with [1], could measure their proposed improvements and we will make sure to include this in our Limitations section of the final version.
> Lastly, the fact that some leaves can have more generative capacity is an intended design choice because it gives the model more freedom to adapt to the data. Having the flexibility to find the right depth for each cluster gives the model more rather than less capabilities.
>
> W3: With respect to the generative capabilities of our work, we agree that it is not state-of-the-art, which is intentional, as this is not the focus of this work. Rather, we want to introduce a new model class that is able to perform generative hierarchical clustering. This is also reflected in our choice of datasets. We want to emphasize that the performance difference to NVAE [3] is not due to the theoretical model formulation but rather due to the chosen architecture and designs.
> We argue that NVAE is in some way a LadderVAE with smart design choices, where we see TreeVAE on the design-choice-level similar to LadderVAE and leave it up to future work, to focus on developing smarter and more involved design choices for our model, which would be an entire work in itself. Thus, we expect that similarly to how TreeVAE outperforms LadderVAE, a version of TreeVAE with highly optimized architecture would outperform NVAE.
>
> W4: Similar arguments to Weakness 3 can be made for the choice of datasets, however, we are currently working on a medical application of our model, as well as are in contact with practitioners in the field of robotics. We decided against the pursuit of including such applications in the paper in order to, given the space constraints, put more focus on the theoretical side of the method. Here, we would also like to note that Newsgroup20 is not an image but a text dataset and that the presented cluster enrichment of CelebA is intended as an exploratory data analysis, rather than for simply rediscovering categorical labels.
>
> Q: In response to your questions, we have conducted an additional experiment, where we a priori fixed the tree structure depicted in Fig. 4 (right) of the submission and trained the full tree without growing. The results are as follows: Acc. $35.1 \pm 4.8$, NMI $54.1 \pm 5.8$, LL $-237.4 \pm 0.4$, RL $228.9 \pm 0.5$, ELBO $241.0 \pm 0.5$. Expectedly, the results are much worse, as a fixed tree is more prone to local optima, providing support of our growing scheme. Regarding Eq. 3 \& Eq. 5, we adopted the notation of LadderVAE (see their Eq. 16) in order to be consistent. For Line 134, the routers of the inference model depend on the intermediate representation $\mathbf{d}_{depth(i)}$ and thus take as input a latent embedding and not $x$. Regarding the type in Eq. 19 \& Eq. 20, thank you for bringing it up, we have changed $z_i$ to $z_i^{(l)}$.
>
> We hope our general response and the response to your review address your concerns. We are happy to address any additional feedback or questions you might have to alleviate your concerns.

---

> > ### Comment · Reviewer_UPHe · 2023-08-12
> > **Thank you for the detailed response!**
> >
> > Having read through the rebuttal and global response, I am willing to change score to recommend acceptance. The authors have address several of my concerns. I still feel that some of the shortcomings in terms of the architectures, datasets and tree-building routine are worth further exploration, but I'm inclined to agree they could be left for future follow-up work. I will be interested to see such extensions!

---

### Author Rebuttal · Authors · 2023-08-09

Dear reviewers,

We deeply appreciate your insightful questions, constructive comments and helpful feedback! Your reviews suggest that you invested a considerable amount of time and effort into understanding our work, for which we are very grateful and thankful. We provide individualized responses to your Weaknesses (W), Questions (Q), and Limitations (L) in separate sections below and hope that our responses address the raised concerns.

Nevertheless, we would like to mention a few points here that might be interesting to everybody:

In the additional PDF, we present an extensive ablation study in Table 1. We have decided a priori to run the rebuttal experiments on Fashion MNIST, as our clustering performance on this dataset is closest to the baselines, therefore, this provides a worst-case analysis for all comparisons. The detail-focused person will observe that the presented values of the default TreeVAE configuration, denoted by "base", are marginally lower(<1%p) than in the paper, which is due to the fact that we trained for slightly shorter, in order to finish the rebuttal in time. All ablation experiments are run over $10$ seeds. Additionally, in Figure 1, we analyze the root embedding of our tree on FMNIST to support why Figure 4 (right) of the main paper clusters bags and trousers together with shoes (for a more detailed discussion, have a look at our response to reviewer bo43). Lastly, Figure 2 shows that the proposed splitting heuristic leads to a stable improvement in the ELBO.

Please find the full bibliography utilized in all responses below.

We are happy to address any further questions or comments from the reviewers to improve the paper and look forward to a fruitful discussion!

[1] "Decision Jungles: Compact and Rich Models for Classification", Shotton et al., 2013

[2] "Twin Contrastive Learning for Online Clustering
", Li et al., 2022

[3] "NVAE: A Deep Hierarchical Variational Autoencoder
", Arash Vahdat and Jan Kautz, 2022

---

### Decision · Program_Chairs · 2023-09-21

**Decision:**

Accept (spotlight)

**Comment:**

All the reviewers liked the paper. As the paper represents a novelty (tree-based hierarchies) in the well-studied variational autoencoder (VAE) field, this AE is of the opinion that the paper deservers a spotlight.